# Synthesis and biological assessment of chalcone and pyrazoline derivatives as novel inhibitor for ELF3-MED23 interaction

Soo-Yeon Hwang[1†], Kyung-Hwa Jeon[1†], Hwa-Jong Lee[1], Inhye Moon[1], Sehyun Jung[1], Seul-Ah Kim[1], Hyunji Jo[1], Seojeong Park[1], Misun Ahn[1], Soo-Yeon Kwak[2], Younghwa Na[2]*, Youngjoo Kwon[1]*

[1]College of Pharmacy and Graduate School of Pharmaceutical Sciences, Ewha Womans University, Seoul, Republic of Korea; [2]College of Pharmacy, CHA University, Pocheon, Republic of Korea

*For correspondence:
yna7315@cha.ac.kr (YN);
ykwon@ewha.ac.kr (YK)

†These authors contributed equally to this work

Competing interest: The authors declare that no competing interests exist.

## eLife assessment

This **valuable** study characterized a new set of small molecules targeting the interaction between ELF3-MED23, with one of the reported compounds representing a promising novel therapeutic strategy, The evidence supporting the conclusions is **convincing**. This article will be of interest to medical and cell biologists working on cancer and, particularly, on HER2-overexpression cancers.

**Abstract** HER2 overexpression significantly contributes to the aggressive nature and recurrent patterns observed in various solid tumors, notably gastric cancers. Trastuzumab, HER2-targeting monoclonal antibody drug, has shown considerable clinical success; however, readily emerging drug resistance emphasizes the pressing need for improved interventions in HER2-overexpressing cancers. To address this, we proposed targeting the protein-protein interaction (PPI) between ELF3 and MED23 as an alternative therapeutic approach to trastuzumab. In this study, we synthesized a total of 26 compounds consisting of 10 chalcones, 7 pyrazoline acetyl, and 9 pyrazoline propionyl derivatives, and evaluated their biological activity as potential ELF3-MED23 PPI inhibitors. Upon systematic analysis, candidate compound **10** was selected due to its potency in downregulating reporter gene activity of *ERBB2* promoter confirmed by SEAP activity and its effect on HER2 protein and mRNA levels. Compound **10** effectively disrupted the binding interface between the ELF3 TAD domain and the 391–582 amino acid region of MED23, resulting in successful inhibition of the ELF3-MED23 PPI. This intervention led to a substantial reduction in HER2 levels and its downstream signals in the HER2-positive gastric cancer cell line. Subsequently, compound **10** induced significant apoptosis and anti-proliferative effects, demonstrating superior in vitro and in vivo anticancer activity overall. We found that the anticancer activity of compound **10** was not only restricted to trastuzumab-sensitive cases, but was also valid for trastuzumab-refractory clones. This suggests its potential as a viable therapeutic option for trastuzumab-resistant gastric cancers. In summary, compound **10** could be a novel alternative therapeutic strategy for HER2-overexpressing cancers, overcoming the limitations of trastuzumab.

## Introduction

Human epidermal growth factor receptor 2 (HER2) belongs to the HER family, encompassing EGFR (HER1, erbB1), HER2 (HER2/neu, erbB2), HER3 (erbB3), and HER4 (erbB4). Known as a proto-oncogene, HER2 plays a pivotal role in various cancer-related processes, including cell migration, proliferation, differentiation, and adhesion (*Iqbal and Iqbal, 2014*; *Gravalos and Jimeno, 2008*; *Kelly and Janjigian, 2016*). Its activation occurs through homo- or hetero-dimerization with other members of the HER family, triggering downstream cellular signaling pathways like Ras-Raf-MAPK and PI3K-AKT-mTOR (*Gravalos and Jimeno, 2008*; *Pellino et al., 2019*). Distinct from other HER family proteins, HER2 does not necessitate ligand-binding for dimer formation or downstream signal activation. Moreover, the structure of the HER2 extracellular domain adopts an open conformation, making it more prone to undergo homo- or heterodimerization. This predisposition of HER2 as a dimerization partner, coupled with its tendency for overexpression, contributes to uncontrolled cell proliferation and the progression of tumors.

Numerous studies have consistently identified HER2 overexpression across diverse cancer types including breast, colorectal, ovarian, pancreatic, prostate, bladder, lung, gastric, and gastroesophageal cancers (*Yan et al., 2015*). The frequency of HER2 overexpression varies: approximately 20% in breast cancers (*Swain et al., 2023*), 5–30% in ovarian cancers (*Chen et al., 2023*), 7–8% in colorectal cancers (*Ivanova et al., 2022*), 0–82% in pancreatic cancers (*Han et al., 2021*), and 7.3–20.2% in gastric and gastroesophageal cancers (*Guan et al., 2023*). In these cases, HER2 status serves as an important prognostic marker, correlating positively with tumor size, invasiveness, and lymph node metastasis (*Gravalos and Jimeno, 2008*; *Swain et al., 2023*; *Ivanova et al., 2022*; *Guan et al., 2023*; *Luo et al., 2018*). The significance of HER2 is evident in the development of separate treatment guidelines tailored specifically for HER2-positive cases in several cancer subtypes such as breast, colorectal, and gastric cancers. This often involves employing trastuzumab, a humanized monoclonal antibody drug targeting the extracellular domain of HER2 protein (*Ajani et al., 2022*; *Gradishar et al., 2020*; *Malla et al., 2023*).

Trastuzumab has significantly improved the clinical outcomes of numerous patients with HER2-positive cancers, but its therapeutic efficacy is considered limited due to readily developing resistance (*Maadi et al., 2021*). The emergence of this persistent refractoriness to trastuzumab has highlighted the need for the establishment of better therapeutic approaches that maintain effectiveness regardless of the development of resistance. While HER2 gene amplification is the primary mechanism responsible for HER2 overexpression in most HER2-positive cancers, except in lung cancer (*Ren et al., 2022*), high transcription rates of HER2 per gene copy have also been observed to contribute (*Liu et al., 2018*). In breast cancer, it has been reported that radiation or endocrine therapy upregulates *ERBB2* gene transcription in HER2-low or HER2-negative subtypes, leading to treatment resistance (*Liu et al., 2018*; *Duru et al., 2012*; *Cao et al., 2009*). Rather than targeting already overexpressed HER2 protein, our focus has shifted to reducing HER2 expression at the genetic level. This involves interrupting the interaction between E74-Like Factor 3 (ELF3) and mediator of RNA polymerase II transcription subunit 23 (MED23) – a transcription factor (TF) and a coactivator for HER2, respectively. As an essential TF for HER2 gene expression, ELF3 directly binds to the ETS transcriptional response element located in the *ERBB2* promoter and interacts with MED23 to drive HER2 overexpression (*Chang et al., 1997*).

Natural and synthetic chalcones have been reported to show a variety of biological efficiency, including anti-proliferative, anticancer, antioxidant, anti-inflammatory, or anti-infective activities (*Liu and Go, 2006*; *Modzelewska et al., 2006*; *Ni et al., 2004*). Chalcone derivatives have shown potential as lead compounds in the field of novel drug discovery due to their promising biological activities and have also been sourced as precursors of flavonoids in higher plants. Licochalcone derivatives, initially identified in *licorice* root, possess alkyl-substituted forms of the typical chalcone structure (*Friis-Møller et al., 2002*), exhibiting various biological activities such as anticancer, antioxidant, anti-inflammatory activities (*Mahapatra et al., 2015*; *Mishra et al., 2001*; *Chen et al., 1997*). Pyrazole, an organic heterocyclic compound, is characterized by a 5-membered ring comprising three carbon atoms and two adjacent nitrogen atoms (*Burgess et al., 2003*). Pyrazoline, an unsaturated pyrazole derivative, contains a single endocyclic double bond on the N-2 position, distinguishing it from pyrazole. Numerous synthetic pyrazoline analogues have demonstrated notable anticancer activities (*Azizur and Anees, 2010*).

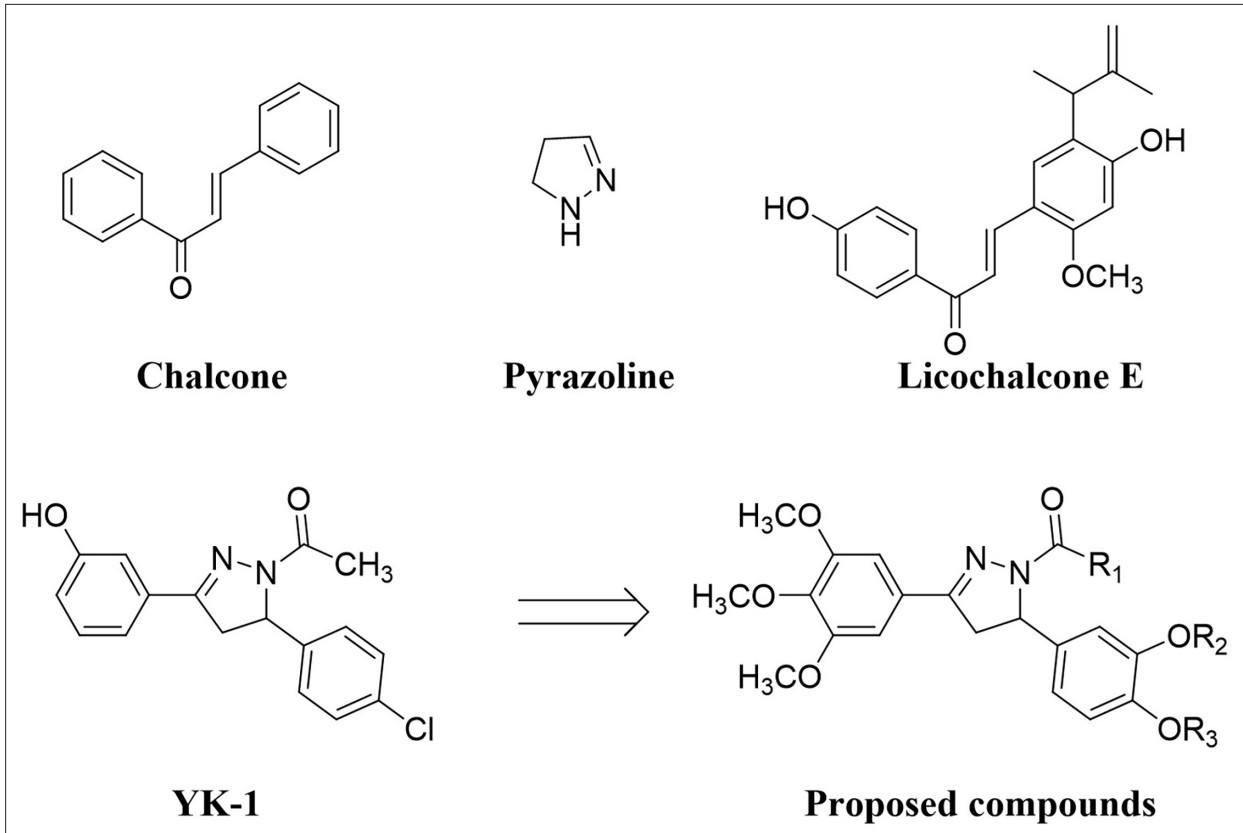

**Figure 1.** Structure of proposed compounds.

In our previous study, pyrazoline analogue YK-1 was developed as an effective inhibitor targeting the interaction between ELF3 and MED23. YK-1 exhibited remarkable efficacy, demonstrating significant in vitro and in vivo anticancer activity specifically against HER2-overexpressing breast and gastric cancers (*Hwang et al., 2023*). As part of our ongoing efforts to discover a potent inhibitor of the ELF3-MED23 PPI for potential anticancer therapies, we designed and synthesized 26 novel compounds, including chalcones, pyrazoline acetyl, and pyrazoline propionyl analogs (*Figure 1*). This exploration led us to elicit a highly potent compound **10** that exhibited excellent in vitro and in vivo anticancer activity against HER2-positive gastric cancer cells.

## Results

### Identification of compound 10 as potent ELF3-MED23 PPI inhibitor

We designed and synthesized 26 novel compounds, including chalcones, pyrazoline acetyl, and pyrazoline propionyl analogs based on the YK1 structure (*Figure 2*). To screen out compounds exhibiting superior inhibitory activity against ELF3-MED23 PPI, we first performed secreted alkaline phosphatase (SEAP) assay (*Hwang et al., 2023*; *Kim et al., 2012*) using 26 synthesized compounds. As in our previous studies, gefitinib and CI-1033 (Canertinib) were used as positive controls for the experiment (*Hwang et al., 2023*; *Kim et al., 2012*; *Figure 3A*). Cell viability tests were conducted in parallel with the SEAP assay under the same experimental condition, in order to confirm that the decrease in fluorescence observed in the SEAP assay was specifically due to ELF3-MED23 disruption, and not related to non-specific cytotoxicity of the compounds (*Figure 3B*). Overall, a total of 12 compounds displayed nearly 100% inhibition at 10 μM, showing the most excellent significant SEAP inhibitory activities among the entire series. Most of these have a chalcone moiety (compounds **1–10** in **group 1**), while compounds **16** and **17** have a pyrazoline acetyl moiety. Among the remaining pyrazoline acetyl derivatives in **group 2**, compounds **11**, **13**, **14**, and **15** also showed inhibitory effects, but their extent was relatively low (compounds **11, 14,** and **15**; below 30% inhibition, compound **13**; 68% inhibition). In

**Figure 2.** Structures of the prepared compounds.

case of the molecules of pyrazoline propionyl skeleton in **group 3**, only compound **26** displayed a moderate inhibitory rate of 63.8%, while majority of them did not show noticeable inhibitory effects (*Figure 3A*). To select the compounds that can actually induce downregulation of HER2 in cancer cells, we applied the top 12 compounds with the highest SEAP inhibitory activity to NCI-N87, a well-known HER2-positive gastric cancer cell line, and evaluated changes in the HER2 protein expression level (*Figure 3C*). Of the tested molecules, only compounds **3**, **5**, and **10** were found to significantly reduce the HER2 levels by more than 50%. However, unlike compound **3** and **10**, the downregulation of HER2 induced by compound **5** did not occur at the mRNA level. This indicates that only compounds **3** and **10** were capable of transcriptionally downregulating HER2 by disrupting the ELF3-MED23 interaction (*Figure 3D*). Since compound **10** remarkably decreased the HER2 level of NCI-N87 by more

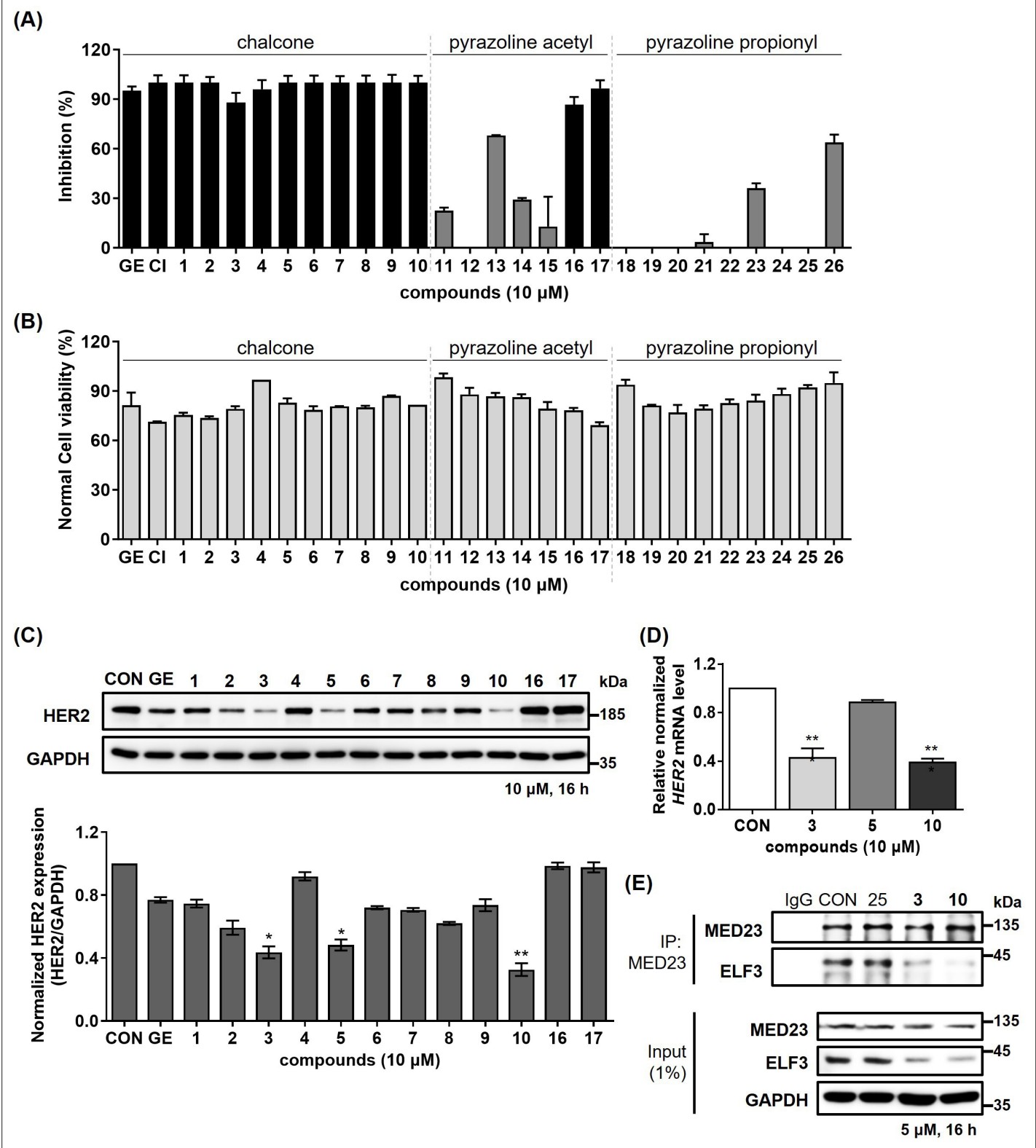

**Figure 3.** Identification of compound 10 as potent ELF3-MED23 PPI inhibitor. (**A**) Synthesized compounds were screened to evaluate their inhibitory activity against ELF3–MED23 PPI. All compounds were used at 10 µM for 12 h (n=3, mean ± S.D.). (**B**) Cell viability was also measured in parallel using the same conditions as for the reporter gene assay in (**A**) (n=3, mean ± S.D.). (**C**) Changes in the HER2 levels were evaluated by treating NCI-N87 cells with the compounds that exhibited high SEAP inhibitory activity in (**A**). (**D**) Effect of compound 3, 5, and 10 on mRNA level of HER2 in NCI-N87 was

*Figure 3 continued on next page*

*Figure 3 continued*

assessed (16 hr treatment at 10 µM, n=3, mean ± S.D., ANOVA, ***p<0.001 *vs.* CON). (**E**) PPI inhibitory activities of compounds **3** and **10** were evaluated against endogenous ELF3 and MED23.Compound **25** was used as negative control.

The online version of this article includes the following source data for figure 3:

**Source data 1.** Raw unedited gels for *Figure 3C*.

**Source data 2.** Uncropped and labelled gels for *Figure 3C*.

**Source data 3.** Raw unedited gels for *Figure 3E*.

**Source data 4.** Uncropped and labelled gels for *Figure 3E*.

effectively disrupting the ELF3-MED23 PPI compared to compound **3** (*Figure 3E*), we finally selected compound **10** as the candidate molecule and performed further experiments.

## Qualitative structure-activity relationship (SAR) of prepared compounds

Field-based approaches primarily focus on characterizing the molecular properties of compounds based on their interaction fields. By providing comprehensive 3D information regarding charge distribution, non-polar interactions, and shape properties of diverse molecules, this method allows detailed insights into how distinct electrostatic, hydrophobic, and steric fields of ligands contribute to their biological activity or inactivity (*Cheeseright et al., 2006*; *Floresta and Abbate, 2021*). Based on this concept, we utilized FieldTemplater software to three-dimensionally align all 26 compounds according to their field points. We subsequently analyzed their configurations through Activity Atlas software (*Tedesco et al., 2024*). We finally visualized the results in 3D activity cliff summary plots of positive/negative electrostatics and favorable/unfavorable hydrophobics (*Figure 4*). According to the electrostatic and hydrophobic plots, the positive/negative and hydrophobic field points of the compounds demonstrating relatively high HER2 inhibitory activity (as exemplified by compounds **3** and **5**) were found to be well-distributed within the favored areas. Especially, the negative field formed around the ketone moiety within the chalcone skeleton, and the hydrophobic field generated by 3-methyl-but-2-enyl side chain of compound **10** are assumed to be important for the HER2 inhibitory effect induced by interrupting ELF3-MED23 PPI. This assumption arises from the observation that the overall activity slightly decreased when both the negative and hydrophobic field point were less distributed in the mentioned regions (*Figure 4A and B*; compounds **3** and **5**). For the compounds displaying low potency (represented by compounds **12**, **15**, **20**, and **24**), their positive/negative electrostatic and hydrophobic potentials shared unfavorable distribution patterns. This suggests that conformation of these compounds is less likely to complementarily fit into the binding interface of ELF3-MED23 PPI (*Figure 4A and B*; compounds **12**, **15**, **20**, and **24**).

## Compound 10 as a transcriptional regulator of HER2 by inhibiting ELF3-MED23 PPI

Previously, we have verified that the TAD domain of ELF3 (129–145 residues; 17 amino acids) interacts with the 391–582 residues of MED23, thereby regulating HER2 at the transcriptional level. Specifically, within ELF3, residues S137 to E144 were identified as essential residues for this interaction (*Hwang et al., 2023*). Based upon the results, we performed in vitro fluorescence polarization (FP) assay by utilizing the $(His)_6$-MED23$_{391-582}$ protein and fluorescein isothiocyanate (FITC)-labeled ELF3$_{129-145}$ peptide (*Figure 4A*). Similar to the positive control, unlabeled ELF3$_{137-144}$ peptide, compound **10** significantly reduced the FP signal induced by the binding of FITC-ELF3$_{129-145}$ and $(His)_6$-MED23$_{391-582}$, demonstrating that compound **10** directly inhibits the ELF3-MED23 PPI. While the $K_i$ value of unlabeled ELF3$_{137-144}$ peptide calculated from the $K_d$ of FITC-ELF3$_{129-145}$ and $(His)_6$-MED23$_{391-582}$ interaction was determined to be 4.47±0.03 µM, compound **10** showed a 6.6-fold lower value of 0.68±0.08 µM, falling within the nanomolar range (*Figure 5A and B*). Through GST-pull down assay, we then confirmed that ELF3-MED23 PPI inhibitory effect of compound **10** is also valid at the cell level (*Figure 5C*). To investigate whether intracellular compound **10** also acts on the binding interface of ELF3 and the 391–582 amino acid region of MED23 within the cell system, we performed split luciferase complementation assay using ELF3- and MED23$_{391-582}$-ligated with N-terminal (Nluc) and C-terminal (Cluc) fragments of split luciferase, respectively (*Figure 5D*). Without compound **10**, ELF3 and MED23$_{391-582}$ successfully

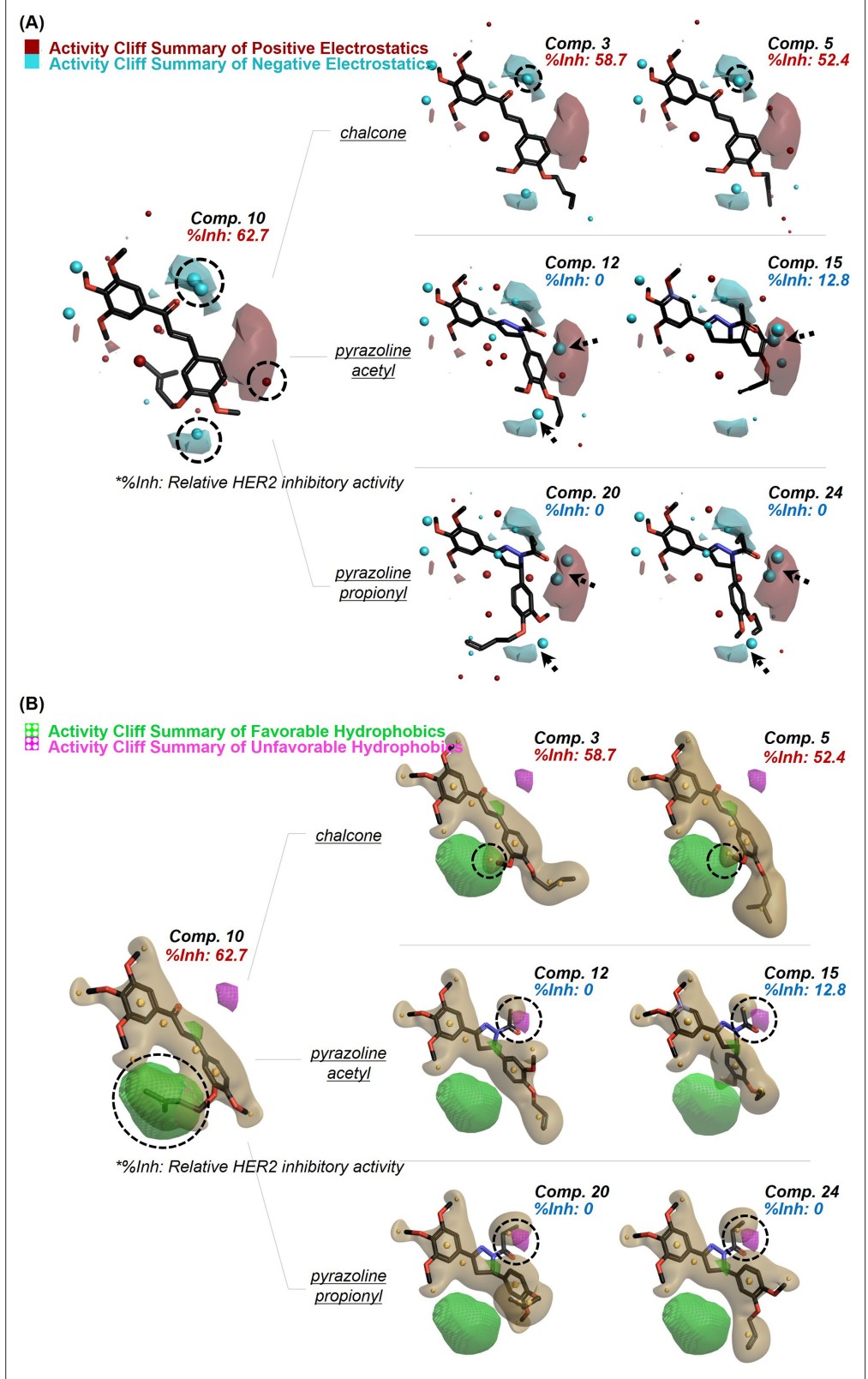

**Figure 4.** Qualitative SAR of prepared compounds. Activity cliff summary for the prepared compound series were plotted regarding the contributions of electrostatic (**A**) and hydrophobic (**B**) fields. The field point pattern was depicted as spheres (Red, positive electrostatics; Cyan, negative electrostatics; Yellow brown, hydrophobic fields).

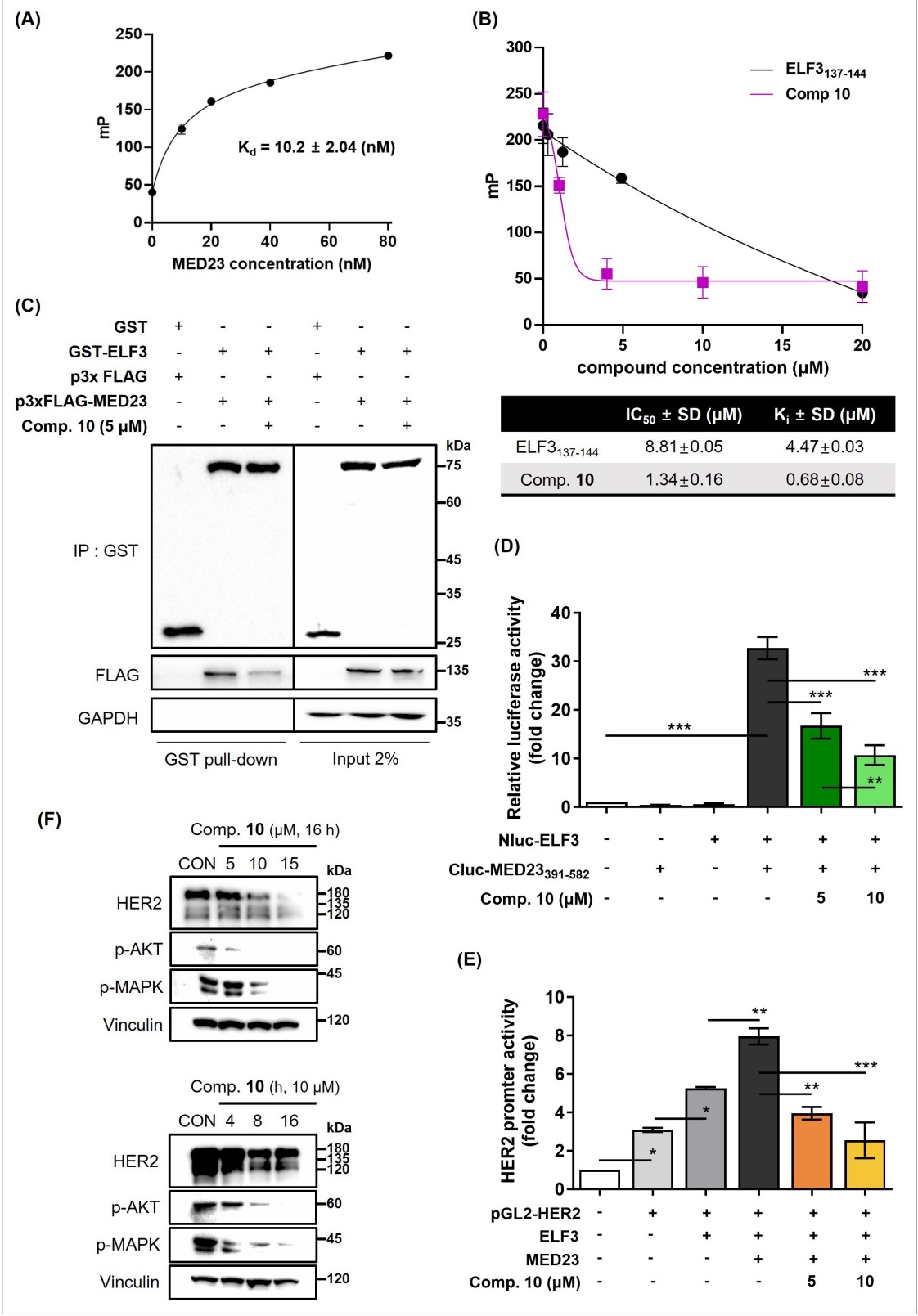

| | IC$_{50}$ ± SD (µM) | K$_i$ ± SD (µM) |
|---|---|---|
| ELF3$_{137-144}$ | 8.81±0.05 | 4.47±0.03 |
| Comp. **10** | 1.34±0.16 | 0.68±0.08 |

**Figure 5.** Compound **10** as a transcriptional regulator of HER2 by inhibiting ELF3-MED23 PPI. (**A**) Titration curve of MED23$_{391-582}$ protein FITC-labeled ELF3$_{129-145}$ peptide. Binding of the MED23$_{391-582}$ protein and ELF3-FITC peptide (17 a.a.) was validated via cell-free FP assay. Kd value was measured as 10.2±0.82 (nM) using the least squares non-linear fit method (n=3, mean ± S.D). (**B**) Effect of compound **10** on the FP (mP) induced by the binding of FITC-ELF3$_{129-145}$ peptide to (His)$_6$-MED23$_{391-582}$ protein was evaluated in cell-free system. Unlabeled ELF3$_{137-144}$ peptide was used as positive control. IC$_{50}$

*Figure 5 continued on next page*

*Figure 5 continued*

and $K_i$ values were calculated from the FP assay results (n=3, mean ±S.D.). (**C**) Intracellular PPI inhibitory effect of compound **10** (5 μM, 12 h treatment) against ELF3-MED23 was evaluated through GST-pull down assay using GST-ELF3$_{WT}$ and 3xFLAG-MED23. (**D**) Impact of compound **10** on the relative luciferase activity generated by Nluc-ELF3$_{WT}$ and Cluc-MED23$_{391-582}$ interaction was evaluated (20 hr treatment at indicated concentrations, n=3, mean ± S.D., ANOVA, ***p<0.001). (**E**) Effect of compound **10** on the overall *ERBB2* promoter activity was assessed (20 hr treatment at indicated concentrations, n=3, mean ± S.D., ANOVA, *p<0.05, **p<0.01, and ***p<0.001). (**F**) Changes in the HER2 and its downstream signaling pathway were evaluated by treating compound **10** in dose- and time-dependent manner.

The online version of this article includes the following source data for figure 5:

**Source data 1.** Raw unedited gels for *Figure 5C*.

**Source data 2.** Uncropped and labeled gels for *Figure 5C*.

**Source data 3.** Raw unedited gels for *Figure 5F*.

**Source data 4.** Uncropped and labelled gels for *Figure 5F*.

interacted within the cell system, subsequently inducing the complementation of Nluc and Cluc to generate a significant amount of bioluminescence (32.7-fold change vs. control). However, compound **10** markedly reduced this luciferase activity in a dose-dependent manner. This result verified that compound **10** is also capable of disrupting the ELF3-MED23 PPI at the cellular level. In fact, the successful inhibition of the ELF3-MED23 PPI by compound **10** ultimately resulted in the attenuation of *ERBB2* promoter activity (*Figure 5E*). This led to a significant downregulation of HER2 expression and its downstream signaling molecules such as pAKT and pMAPK in a concentration- and time-dependent manner (*Figure 5F*).

## Compound 10 as a potent anticancer agent for HER2-positive gastric cancer cells

Based upon the obtained results so far, we proceeded to evaluate whether compound **10**-mediated inhibition of ELF3-MED23 PPI could lead to significant anticancer activity against HER2-positive gastric cancer cells in both in vitro and in vivo settings. First, by treating NCI-N87 cells with varying concentrations of compound **10**, we found that apoptosis was significantly induced in a concentration-dependent manner (*Figure 6A*). This effect was further validated by the dose- and time-dependent increases in pro-apoptotic markers, cleaved PARP (c-PARP) and cleaved caspase 3 (c-Caspase 3) (*Figure 6B*). Accordingly, compound **10** significantly decreased the cell proliferation of NCI-N87 in a concentration-dependent manner (*Figure 6C*), indicating its potent anticancer activity in the in vitro system. To evaluate its potency in the in vivo setting, we then established an NCI-N87 xenograft mouse model and administered compound **10** intravenously. Notably, the administration of 4 mg/kg of compound **10** resulted in significant retardation of tumor growth (*Figure 6D*), leading to a marked reduction in the final tumor volume, approximately 3.5-fold decrease compared to the control group (*Figure 6E and F*). Through Immunohistochemistry (IHC) analysis, we confirmed that the proliferation marker, Ki67, was remarkably downregulated (a 2.3-fold decrease compared to the control) in the tumor tissues administered with compound **10** (*Figure 6G*, right panel). This reduction in Ki67 expression was accompanied by a significant decrease in HER2 levels in the tumor tissue (a 1.7-fold decrease compared to the control), suggesting that the overall anti-tumor activity of compound **10** is primarily due to its capacity to inhibit HER2 expression (*Figure 6G*, left panel).

## Compound 10 as a novel strategy to overcome trastuzumab resistance

As previously mentioned, trastuzumab has been a primary first-line therapy for various HER2-overexpressing cancer subtypes, including gastric cancer (*NCCN, 2019*; *Gradishar et al., 2020*). However, one of the significant drawback of this drug is the development of resistance, which readily occurs within 1 year of medication (*Fiszman and Jasnis, 2011*). In our previous study, we demonstrated that inhibiting the ELF3-MED23 PPI to downregulate HER2 at the transcriptional level could serve as a potentially promising alternative therapeutic option to trastuzumab (*Hwang et al., 2023*). Consequently, we sought to determine whether compound **10** holds the capability to overcome trastuzumab resistance. For this assessment, we utilized the trastuzumab-refractory NCI-N87 (NCI-N87 TR) cell line, which exhibited a 5.3-fold lower response rate to 10 μg/mL trastuzumab compared to the parental NCI-N87 cells (Parent *vs.* TR; 50.1% *vs* 9.5% growth inhibition; *Figure 7A*). At a concentration

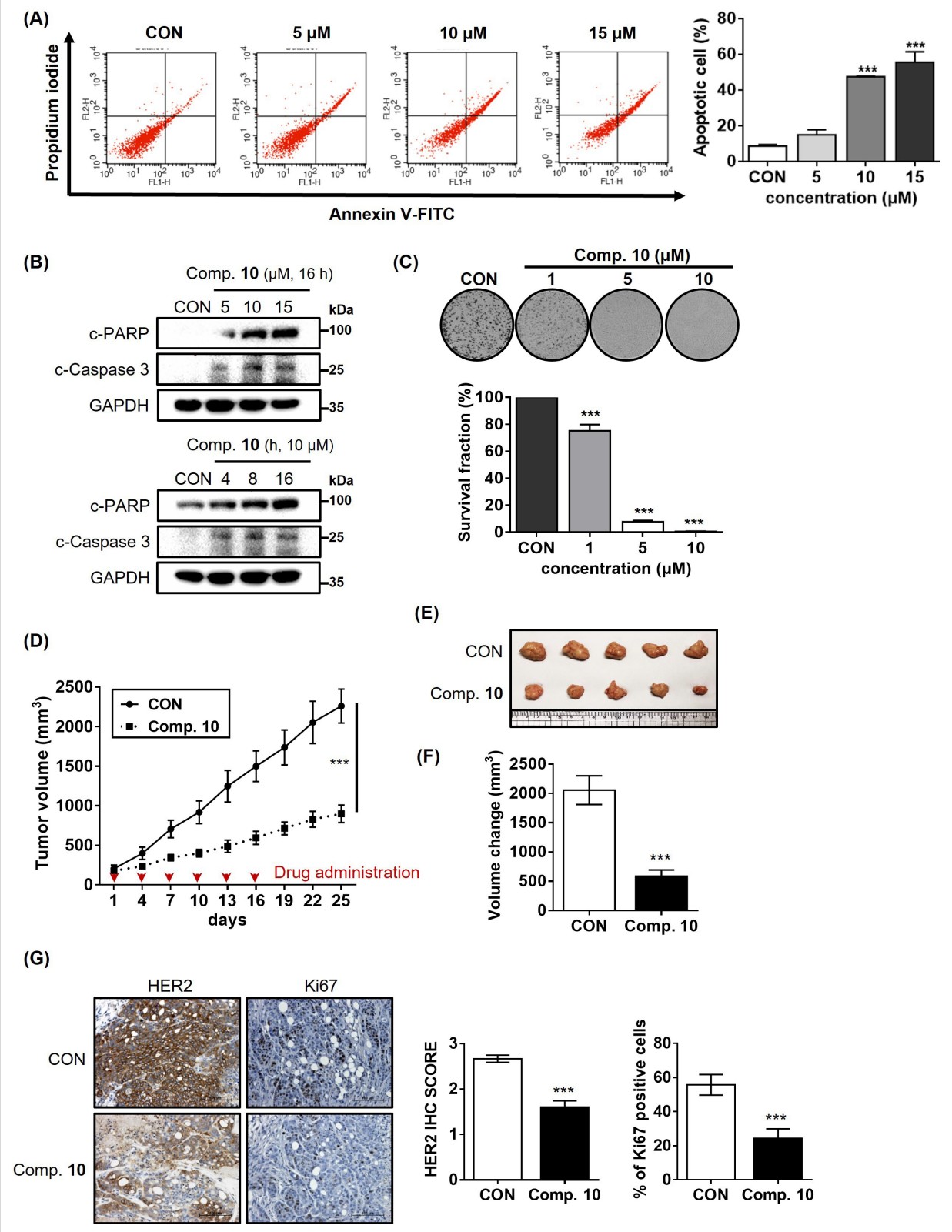

**Figure 6.** Compound 10 as a potent anticancer agent for HER2-positive gastric cancer cells. (**A**) Apoptosis induced by compound 10 in parental NCI-N87 was assessed by treatment with the compound in a dose-dependent manner (24 hr treatment at indicated concentrations, n=3, mean ± S.D., ANOVA, ns = non-significant, ***p<0.001 vs. CON). (**B**) Changes in the pro-apoptotic markers were evaluated by treating compound 10 in dose- and time-dependent manner. (**C**) Anti-proliferative effect of compound 10 (10 µM) was evaluated against NCI-N87 cell line (10 days of treatment at indicated

*Figure 6 continued on next page*

*Figure 6 continued*

concentrations, n=3, mean ± S.D., ANOVA, ***p<0.001 vs. CON). (**D**) Tumor growth inhibitory effect of compound **10** was evaluated using NCI-N87 xenograft mouse model (n=5 per group; intravenous (IV) injection at 4 mg/kg every 3 days, indicated by red arrows). Tumor volumes were evaluated at the indicated time points by measuring the length and width of the tumor with calipers using the equation (length x width$^2$)/2. Data was indicated as mean ± S.E.M. (**E**) Photograph of the tumors collected from the vehicle- and compound **10**-treated mice. (**F**) Final volume changes were assessed for the tumors excised from each experimental group (n=5, mean ± S.E.M., Student's t-test, ***p<0.001 vs. CON) (**G**) IHC analysis was conducted against HER2 and Ki67 in the tumors (scale bar = 100 μm). Score quantification was performed using Image J software (10 independent fields per sample were evaluated, mean ± S.D., Student's t test, ***p<0.001 vs. CON).

The online version of this article includes the following source data for figure 6:

**Source data 1.** Raw unedited gels for *Figure 6B*.

**Source data 2.** Uncropped and labelled gels for *Figure 6B*.

---

of 10 μg/mL, trastuzumab exhibited no discernible impact on the level of HER2 or its downstream signaling molecules such as p-AKT and p-MAPK in NCI-N87 TR cells. In contrast, compound **10** effectively attenuated these pathways by inhibiting HER2 expression (*Figure 7B*). Accordingly, compound **10** markedly inhibited the long-term proliferation of NCI-N87 TR, unlike trastuzumab (*Figure 7C*). Moreover, the dose-dependent treatment of compound **10** significantly induced apoptosis in NCI-N87 TR cells (*Figure 7D*). These findings collectively suggest that compound **10** demonstrates anticancer

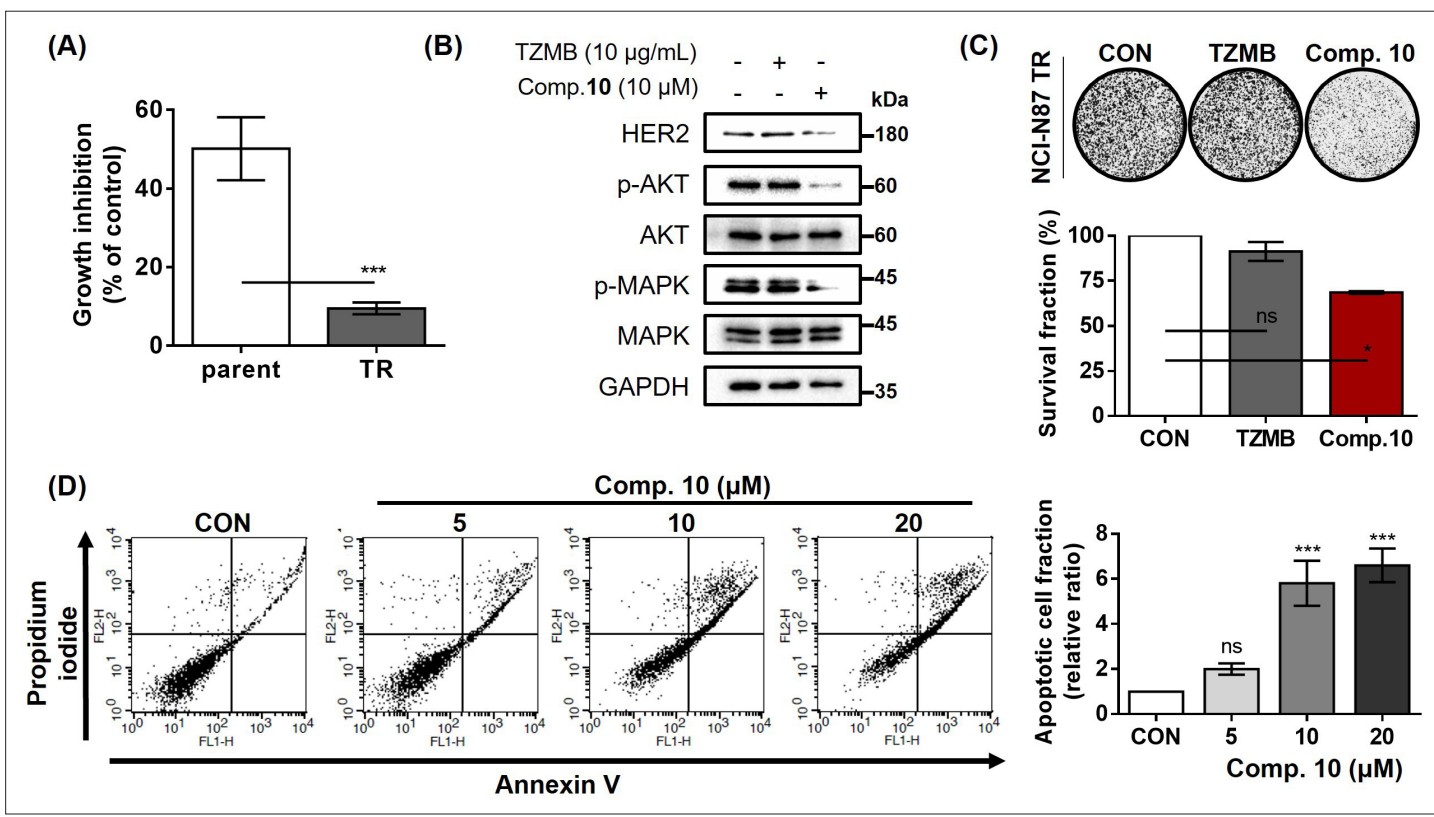

**Figure 7.** Compound 10 as a novel strategy to overcome trastuzumab resistance. (**A**) Trastuzumab (10 μg/ml, 24 hr)-induced growth inhibitory effects were assessed against parental and TR NCI-N87 cell lines (n=3, mean ± S.D., Student's t-test, ***p<0.001 *vs.* parent). (**B**) Trastuzumab- and compound 10-mediated changes of HER2 and its downstream signaling pathway were evaluated (16 hr treatment at indicated concentrations). (**C**) Anti-proliferative effects of trastuzumab (10 μg/mL) and compound 10 (10 μM) were evaluated against NCI-N87 TR cell line. Each compound was treated for 10 days (n=3, mean ± S.D., ANOVA, ns = non-significant, *p<0.05 *vs.* CON). (**D**) Apoptosis induced by compound 10 in NCI-N87 TR was assessed by FACS analysis with treatment of the compound in a dose-dependent manner (24 hr treatment at indicated concentrations, n=3, mean ± S.D., ANOVA, ns = non-significant, ***p<0.001 *vs.* CON).

The online version of this article includes the following source data for figure 7:

**Source data 1.** Raw unedited gels for *Figure 7B*.

**Source data 2.** Uncropped and labelled gels for *Figure 7B*.

activity not only against parental HER2-positive cancer cells but also against their trastuzumab-refractory clones.

## Discussion

As HER2-positivity is generally responsible for more aggressive behavior of diverse cancer subtypes, HER2 has long been regarded as a crucial biomarker and a primary therapeutic target. Consequently, numerous researchers have consistently sought to develop new HER2-targeting agents beyond trastuzumab, and indeed, several drugs such as Enhertu (fam-trastuzumab deruxtecan-nxki) and Tukysa (tucatinib) have been identified and approved by the FDA in recent years (*Ferrando-Díez et al., 2022*; *Tarantino et al., 2021*).

To date, numerous HER2-targeting drugs are clinically available. Among them, trastuzumab, the first FDA-approved HER2-targeting drug, still remains the most commonly used drug of molecular targeted therapy for HER2-positive breast cancer, improving the survival of patients with this molecular subtype (*Kreutzfeldt et al., 2020*; *Wang et al., 2022*). However, despite initial responses to trastuzumab or trastuzumab conjugate regimens, approximately 70% of HER2-positivie breast cancer patients develop resistance within the first year of treatment (*Kreutzfeldt et al., 2020*; *Wang et al., 2022*; *Vivekanandhan and Knutson, 2022*). This high incidence of trastuzumab resistance highlights the persistent challenges of relapse and drug resistance in the treatment of HER2-positive breast cancer.

As HER2 targeting primarily involves the already overexpressed HER2 and its downstream signaling pathways, typically driven by the gene amplification (*Cresti et al., 2016*), alternative approaches have been explored to downregulate HER2 signaling from the gene expression level. Specifically, strategies have been investigated to inhibit the interaction between transcription factor (TF), ELF3, and its coactivator, MED23 (*Hwang et al., 2023*; *Kim et al., 2012*; *Nam et al., 2013*; *Shimogawa et al., 2004*; *Li et al., 2021*). ELF3, a critical regulator of HER2 gene expression, acts on the ETS transcriptional response element of *ERBB2* promoter and interacts with MED23 to promote HER2 overexpression (*Chang et al., 1997*; *Asada et al., 2002*). Our previous work has provided structural insights into the regulation of the ELF3-MED23 PPI (*Hwang et al., 2023*). Expanding on the findings that interrupting the ELF3-MED23 PPI can effectively serve as a viable alternative strategy for inhibiting HER2 compared to trastuzumab, we designed a series of potential ELF3-MED23 inhibitors incorporating chalcone and pyrazoline moieties.

This study primarily aims to identify novel small molecules that can inhibit HER2 with outstanding anticancer activity, which can be broadly applicable to diverse HER2-positive cancers, irrespective of their resistance to trastuzumab. Following a comprehensive biological evaluation, we successfully identified compound **10,** demonstrating a highly potent ELF3-MED23 inhibitory efficacy ($K_i$ = 0.68 ± 0.08 μM). Based on these findings, we confirmed that this compound exhibits excellent anticancer activity in both in vitro and in vivo settings against HER2-overexpressing gastric cancer cells and those exhibiting resistance to trastuzumab.

This study highlights that the chalcone moiety can efficiently bind to the ELF3-MED23 interface, suggesting that this structure can potentially serve as a useful scaffold in the development of PPI inhibitors, specifically for downregulating HER2 and its associated signaling cascade. The results also clearly demonstrates that inhibition of ELF3-MED23 PPI through the chalcone derivative can serve as a potential therapeutic strategy targeting HER2 in anticancer treatments. By combining our findings on the efficacy of chalcone derivatives with our understanding of the complexity of the ELF3-MED23 interaction and its role in HER2 regulation, this study makes a valuable contribution, paving the way for designing targeted therapies. Consequently, we anticipate that these insights will provide valuable contributions toward development of new agents, the progression of treatments, and the enhancement of the therapeutic landscape for HER2-positive cancers.

## Materials and methods
### Chemistry

The synthesized compounds were classified into three groups: **group 1** encompassing chalcone compounds, **group 2** comprising *N*-1-acetyl-4,5-dihydropyrazoline compounds, and **group 3**

consisting of *N*-1-propionyl-4,5-dihydropyrazoline compounds (*Scheme 1* and *Figure 2*). **Group 1** chalcone compounds were synthesized through a modified *Claisen-Schmidt* condensation reaction using 3,4,5-trimethoxyacetohenone and the appropriate aromatic aldehyde under 50% NaOH base condition, yielding between 47.8% to 88.4%. Analysis through [1]H-NMR spectroscopy confirmed the trans conformation of chalcone compounds, evidenced by two doublet peaks with coupling constants around 15.6 Hz, representing α and β hydrogens in the conjugated ketone structure. **Group 2** acetyl-pyrazoline compounds were obtained by reacting **group 1** chalcone compounds with hydrazine and acetic acid, yielding between 50.8% to 82.4%. [1]H-NMR spectra of **group 2** compounds displayed characteristic peaks of three double of doublets at 3.1, 3.7, and 5.5 ppm, indicating each hydrogen at *C*-4 and *C*-5 positions in the pyrazoline ring structure. Compounds **11–17** also showed a methyl peak as a singlet at about 2.4 ppm, corresponding to the acetyl group of *N*-1 position. Interestingly, during this pyrazoline ring construction process, chalcone compounds **5** and **10** yielded compounds **11** and **16** with the *O*-substituted 3-methyl-but-2-enyl moiety deleted instead of the desired acetyl compounds. **Group 3** propionyl-pyrazoline compounds were also prepared in the same manner as **group 2** compounds with yields of 4.9% to 79.7%, except for propionic acid conditions. [1]H-NMR spectra confirmed the presence of a propionyl substituent at the *N*-1 position of the pyrazoline ring, displaying a set of triplet and double of quartet peaks at about 1.2 and 2.7 ppm, respectively, corresponding to the ethyl peak of the propionyl group. All synthesized compounds are consistent with the spectral data.

**Scheme 1.** General synthesis methods for the target compounds.

Chemicals and reagents used were obtained from Aldrich Chemical Co. and others were from company like TCI. Melting points were measured without correction in open capillaries with Barnstead Electrothermal melting point apparatus, Manual MELTEMP (Model No: 1202D). Chromatographic separations were monitored by thin-layer chromatography using a commercially available pre-coated Merck Kieselgel 60 F254 plate (0.25 mm) and detected by visualizing under UV at 254 and 365 nm. Silicagel column chromatography was carried out with Merck Kieselgel 60 (0.040–0.063 mm). All solvents used for chromatography were directly used without distillation. The purity was assessed by HPLC (Shimadzu LC-20AD) analysis under the following conditions; column, SunFire C18 (4.6 mm × 150 mm, 5 mm); mobile phase, condition: A (water) and B (acetonitrile) using a linear gradient of 50–70% B in 0–15 min, 70% B in 15–20 min, 100% B in 20–25 min and 50% B in 25–30 min, flow rate; 1.0 mL/min; detection, diode array detector (Shimadzu Spd-M20A). The purity of compound is described as percent (%) and retention time was given in minutes. NMR spectra were recorded on Varian AS 400 ([1]H NMR at 400 MHz and [13]C NMR at 100 MHz) with tetramethylsilane as an internal standard (*Supplementary file 1*). Chemical shift (d) values are expressed in ppm and coupling constant (J) values in hertz (Hz). The melting points were measured on Gallenkamp Melting Point Apparatus without correction. Detailed methods for synthesis of each compound were indicated in Appendix 1.

**Table 1.** Summary of the antibody lists used in this study.

| Antibody | Vendor | Catalogue number | Application | Dilution |
|---|---|---|---|---|
| AKT | Cell signaling | #9272 | WB | 1:2000 |
| c-PARP | Cell signaling | #9541 | WB | 1:2000 |
| ELF3 | Thermo Fisher | PA5-21293 | WB | 1:2000 |
| FLAG | MBL | M185-3L | WB | 1:1000 |
| GAPDH | MBL | M171-3 | WB | 1:10000 |
| GST | MBL | M209-3 | WB | 1:1000 |
|  |  |  | WB | 1:1000 |
| HER2 | Thermo Fisher | MA5-13105 | IHC | 1:100 |
| Ki67 | Dako | M7240 | IHC | 1:200 |
| MAPK | Cell signaling | #9102 | WB | 1:2000 |
| MED23 | Novus | NB200-338 | WB | 1:2000 |
| p-AKT (S473) | Santa cruz | sc-7985 | WB | 1:2000 |
| p-MAPK (T202/Y204) | Cell signaling | #9101 | WB | 1:2000 |
| Cleaved caspase-3 | Cell signaling | #9661 | WB | 1:1000 |

## Cell culture

NCI-N87 cell line was cultured at 37 °C in a humidified atmosphere with 5% $CO_2$. NCI-N87 cells were obtained from Korean Cell Line Bank (KCLB No. 60113) and cultured in RPMI (Welgene, Korea) containing 10% fetal bovine serum (FBS, Hyclone, USA).

## Secreted alkaline phosphatase (SEAP) assay

SEAP assay was performed as previously described to measure ELF3-MED23 PPI-dependent HER2 transcription (*Hwang et al., 2023*). In this assay, the GAL4-ELF3 fusion protein binds to one of the five GAL4 binding sites on the reporter gene (pG4IL2SX). The interaction between the GAL4-ELF3 fusion protein and endogenous MED23 induces the expression of the SEAP. Once expressed, SEAP acts as a phosphatase on the substrate 4-MUP (4-methyl umbelliferyl phosphate), resulting in increased fluorescence. The mammalian expression vector, pBJ-GAL4-ELF3 was co-transfected with the reporter gene, pG5IL2SX to 293T human kidney cells. Compounds were treated for 16 hr, and each of the cultured medium was analyzed to measure the changes in SEAP activity.

## Western blot analysis

Western blot analysis was performed as described previously (*Hwang et al., 2023*). For the experiment, indicated cells were seeded in 6-well plates and incubated until they reached ~70% confluence.

**Table 2.** Information of utilized qRT-PCR primers.

| Gene | | Sequence |
|---|---|---|
| Actin | Forward | 5' AGCCATGTACGTAGCCATCC 3' |
| | Reverse | 5' CTCTCAGCTGTGGTGGTGAA 3' |
| ERBB2 | Forward | 5' GGTGGTCTTTGGGATCCTCA 3' |
| | Reverse | 5' ACCTTCACCTTCCTCAGCTC 3' |
| ELF3 | Forward | 5' GTGATGCTGAGCTTGGGATG 3' |
| | Reverse | 5' TTAGGTTAGAAGCGCCCACA 3' |
| GAPDH | Forward | 5' GAGTCAACGGATTTGGTCGT 3' |
| | Reverse | 5' GACAAGCTTCCCGTTCTCAG 3' |

Stock solutions were prepared in DMSO and PBS for compound **10** and trastuzumab, respectively. The stock solutions were individually treated with varying volumes to meet the final concentration as indicated. Media were exchanged to serum-free media right before the compound treatment. All of the antibodies utilized in this study are summarized in *Table 1*.

### RNA extraction and quantitative real-time PCR

Cells were seeded in 60 mm cell culture dishes at a density of $5\times10^5$ cells per dish and incubated overnight. The cells were washed with PBS and treated with serum-free medium containing compounds at indicated concentrations. Total RNA from the compound treated cells was extracted using the Tri-RNA Reagent (FAVORGEN Biotech Corp., Taiwan) and cDNA was synthesized from the extracted RNA using PrimeScriptTM RT Reagent Kit (Takara Bio Inc, Japan) according to the manufacturer's instruction. Quantitative analysis of the *ERBB2* was performed using a SensiFAST SYBR Hi-ROX kit (Bioline, Canada). PCR amplification condition was applied as described previously (*Hwang et al., 2023*). Relative quantity of mRNA was calculated using the ΔΔCt method and normalized by *GAPDH*. The primer sequences used in this study are summarized in *Table 2*.

### Co-immunoprecipitation (Co-IP) assay

NCI-N87 cells were seeded in 100 mm cell culture dish. When they reached to 70~80% confluency, 10 µM of each candidate compounds were treated for 16 hr. Cell lysates were prepared under the same procedure utilized in western blot analysis. 400 µg of each protein aliquots were incubated with MED23 antibody or rabbit-IgG antibody for 6 hr under rotary agitation at 4 °C. Rabbit-IgG-treated samples were compared as control. Fifteen µl of the protein-A/G PLUS-agarose beads (sc-2003, Santa Cruz, CA) were then added to each the samples and again rotated at 4 °C for 8 hr. The incubated samples were centrifuged for 20 min under 13,000 rpm. After discarding the supernatant, the beads were washed with 200 µl lysis buffer for three times. Fifteen µl of $2\mathrm{X}$ loading buffer (1 M Tris-HCl (pH6.8), glycerol, 10% SDS, $H_2O$, bromophenol blue, 2-mercaptoethanol) was added right after removing the supernatant from last washing step, and boiled at 100 °C for 5 min to denature the protein and detach from the beads. The samples were loaded on SDS-PAGE after centrifugation at 1300 rpm for 10 min. The following western blot protocol was same as above.

### Fluorescence polarization (FP) assay

FP assay was conducted following a previously described method to evaluate the molecular interaction between ELF3 and MED23 (*Hwang et al., 2023*). The FP assay operates on the principle of the molecular rotation dynamics. When a fluorescently labeled small molecule is excited by polarized light, the fluorescence emitted can be either polarized or depolarized depending on the molecular status. Free small molecules rotate rapidly, altering the orientation of their fluorescence dipole and emitting depolarized light. However, when these small moelcules bind to large molecules such as proteins, the resulting complex rotates more slowly, and the emitted light retains much of its original polarization. In this study, different concentrations of $(His)_6$-MED23$_{391-582}$, as the large molecule, and 10 nM of FITC-labeled ELF3$_{129-145}$ peptide, as the fluorescence-labeled small molecule, were combined in assay buffer to determine the $K_d$ value in advance. The calculations were performed using Prism 6.0 (GraphPad Software, USA) with the least-squares non-linear fit method. The FP signals were measured in millipolarization (mP) units using the Infinite F200 PRO microplate reader (Tecan Group Ltd., Switzerland) at excitation/emission wavelengths of 485/535 nm. For the displacement assays, varying concentrations of compound **10** and unlabeled ELF3$_{137-144}$ peptide were separately added to the mixture containing 80 nM of $(His)_6$-MED23$_{391-582}$ and 10 nM of FITC-ELF3$_{129-145}$. Activity of compound **10** was assessed by determining the IC$_{50}$ of each compound, using the four-parameter logistic equation with Table Curve 2D program (SPSS Inc). Finally, the $K_i$ values were calculated based on the Cheng-Prusoff equation: $K_i = IC_{50}/1+([Ligand]/K_d)$.

### Glutathione S-transferase (GST) pull-down assay

Indicated plasmids were transduced to HEK293 cell using JetPRIME (Polyplus transfection, France). Five µM of compound **10** was co-applied to the system. Cell lysis was prepared under the same procedure as for the western blot analyses (*Hwang et al., 2023*). A total of 1000 µg of cell lysates were incubated overnight with Glutathione Sepharose beads (GE Healthcare, UK) at 4 °C on a rotator. Beads

were washed three times with ice-cold 1 x PBS, and eluted with elution buffer (20 mM Glutathione, 100 mM Tris-HCl (pH 8.0), 120 mM NaCl, 10% glycerol). The precipitated proteins were analyzed via western blot.

## Split-luciferase complementation assay

All split luciferase biosensors were made by In-Fusion HD Cloning kit using firefly luciferase from the pGL3-basic vector (Promega, USA) template. Luciferase was split into two fragments (Nluc and Cluc). Full-length cDNA for ELF3 was fused with the N-terminal fragments of the split luciferase (Nluc), and the MED23$_{391-582}$ fragment was fused with the C-terminal fragments of the split luciferase (Cluc). Both fragments were cloned into the p3Xflag-myc-CMV26 vector. The HEK293 cells were co-transfected with Nluc-ELF3 and Cluc-MED23$_{391-582}$. After incubation for 6 hr, the transfected HEK293 cells were washed with PBS and treated with compound in a serum-free medium for 20 hr. HT59 was treated at different concentrations.

## Luciferase promoter assay

HEK293 cells were plated in 60 mm culture dishes and transfected with 1 μg of pGL2-HER2 alone or in combination with 0.5 μg of pcDNA3.1-flag-ELF3 and p3Xflag-myc-CMV26-MED23. All transfections were conducted using jetPRIME Transfection Reagent (Polyplus-transfection, FRANCE). After incubation for 6 hr, the culture medium was replaced with serum-free medium containing different concentrations of HT59 for 20 hr. After 20 hr, luciferase activities were measured with the Tecan Infinite 200 PRO microplate reader (Tecan, Tecan Group Ltd., Switzerland) using the Luciferase Assay system (Promega, USA), according to the manufacturer's protocols.

## Annexin V/PI double staining apoptosis assay

NCI-N87 cells were seeded in 60 mm dishes at a density of $5\times10^5$ cells per dish. When cells reached 80% confluence, the cells were treated with 5, 10, and 20 μM of compound **10** for 16 hr. Cells were washed with PBS and harvested using Trypsin-EDTA and centrifugation at 3200 rpm for 3 min. FITC-Annexin V apoptosis detection kit 1 (BD Pharmingen) was used to evaluate compound-induced apoptosis. Pellets were washed with PBS and incubated with 100 μL of 1×Annexin V binding buffer containing propidium iodide and FITC-Annexin V for 20 min in the dark at room temperature. The samples were diluted by adding 400 μL of 1×Annexin V binding buffer and then analyzed using Fluorescence-Activated Cell Sorting (FACS; BD Biosciences, USA). At least 10,000 cells were measured for each sample.

## Clonogenic assay

Cells were seeded at a density of 1000 cells per well in six-well plates and incubated overnight. After the cells were treated with serum-free medium containing compounds at the indicated concentrations for 24 hr, the medium was replaced to growth medium without compounds. The media was changed every 3 days. The plates were incubated for 9 days to allow colony formation. The colonies were washed with PBS, fixed with 100% methanol for 3 min at room temperature, and then stained with crystal violet (0.5% in 100% Methanol) for 10 min. The images were obtained using ChemiDoc (bio-image analyzer, BR179-8280) and the number of colony was counted using the image J program.

## Tumor xenografts

NCI-N87 cells ($5\times10^6$ cells) were implanted into the flank of 5-week-old female athymic nude mice (Envigo, USA) using 100 μL of 1 x PBS. The mice were then assigned randomly to different groups once the average tumor volume reached 90–100 mm$^3$. Subsequently, the drug was administered intravenously to NCI-N87 xenografts, with six repetitions at 3 day intervals. Compound **10** was given at the concentration of 4 mg/kg, prepared in DMAC/Tween80/saline (5:10:85) mixture along with saline. Tumor size changes were monitored for additional days after the last drug injection until the average tumor size of the control group reached 2000–2500 mm$^3$. Mice were sacrificed 25 days after the first drug injection, and tumors were immediately excised from each mouse. The relative tumor sizes were determined by measuring the tumor length (L) and width (W) with calipers and the formula (L × W$^2$) / 2 was used for calculations. Animal handling was conducted in accordance with ethical guidelines

approved by the Animal Experiment Ethics Committee of Ewha Womans University, adhering to relevant regulations (IACUC20-008).

## IHC assay for xenograft mouse model

Tumors excised from the xenograft mouse model were processed into paraffin-embedded block sections for IHC. The experiments were carried out following standard protocols. The specified HER2 and Ki67 primary antibodies were incubated at 4 °C overnight and then washed 3 times with 1 x PBS. Subsequently, they were exposed to secondary antibodies, developed using a Vectastain ABC kit (Vector Laboratories, USA), and stained with DAB solution (Dako, Carpinteria, USA), all in accordance with the manufacturers' protocols. The IHC staining was further counterstained with hematoxylin (USA), and was evaluated under light microscopy at ×200 magnification. A semi-quantitative IHC score was adopted for evaluation, where the final scores were calculated by multiplying the intensity and fraction score (percentage of samples counted at each scale), leading to a range from 0 to 300. All imaging and evaluation were conducted using an Axiophot 2 apparatus (Carl Zeiss MicroImaging Inc, Thornwood, NY, USA), at the Drug Development Research Core Center. The details of the applied antibodies and dilution ratios can be found in *Table 1*.

## In silico qualitative SAR analysis

For the qualitative SAR analysis Cresset's Flare v7.1 software was utilized. All the prepared 28 compounds were primarily aligned in FieldTemplater to generate conformations and molecular fields for each compound. Using compound 10 as the reference molecule, qualitative Activity Atlas model was subsequently constructed under the default settings *Haider et al., 2022* to visualize the activity cliff summary of electrostatics and hydrophobics in 3D plots.

## Development of trastuzumab resistant NCI-N87

To develop trastuzumab-resistance NCI-N87 cells, NCI-N87 cells were continuously treated with gradually increased amount of trastuzumab. Cells were cultured in medium containing 3–10 μg/ml of trastuzumab until 30th passage.

## Statistical analysis

All experiments were performed at least three times and all results were expressed as mean ± standard deviation. Statistics were analyzed via one-way ANOVA or Student's t-test with Prism 6.0 (GraphPad Software, USA). Differences between two values were considered statistically significant when the p-values (demonstrated as single, double, or triple asterisks) were <0.05, <0.01, and <0.001.

## Acknowledgements

This work was supported by grants from the National Research Foundation of Korea (NRF), funded by the Korean government (MSIT) (NRF-2018R1A5A2025286, 2021M3E5E7024855, 2022R1A2C2092053), and by a grant from the Korea Basic Science Institute (National Research Facilities and Equipment Center), funded by the Ministry of Education (2021R1A6C101A442).

## Additional information

### Funding

| Funder | Grant reference number | Author |
| --- | --- | --- |
| National Research Foundation of Korea | NRF-2018R1A5A2025286 | Youngjoo Kwon |
| National Research Foundation of Korea | 2021M3E5E7024855 | Youngjoo Kwon |
| National Research Foundation of Korea | 2022R1A2C2092053 | Youngjoo Kwon |

| Funder | Grant reference number | Author |
| --- | --- | --- |
| Korea Basic Science Institute | 2021R1A6C101A442 | Youngjoo Kwon |

The funders had no role in study design, data collection and interpretation, or the decision to submit the work for publication.

## Author contributions

Soo-Yeon Hwang, Data curation, Investigation, Visualization, Writing - original draft; Kyung-Hwa Jeon, Data curation, Investigation, Visualization, Writing - review and editing; Hwa-Jong Lee, Inhye Moon, Sehyun Jung, Seul-Ah Kim, Hyunji Jo, Seojeong Park, Misun Ahn, Soo-Yeon Kwak, Investigation; Younghwa Na, Conceptualization, Supervision, Investigation, Writing - review and editing; Youngjoo Kwon, Conceptualization, Supervision, Funding acquisition, Project administration, Writing - review and editing

## Author ORCIDs

Youngjoo Kwon ⓘ https://orcid.org/0000-0001-6256-3042

## Ethics

Animal handling was conducted in accordance with ethical guidelines approved by the Animal Experiment Ethics Committee of Ewha Womans University, adhering to relevant regulations (Permit Number: IACUC20-008). Every effort was mde to minimize suffering during experiments.

Reviewer #1 (Public review): https://doi.org/10.7554/eLife.97051.3.sa1
Reviewer #2 (Public review): https://doi.org/10.7554/eLife.97051.3.sa2
Reviewer #3 (Public review): https://doi.org/10.7554/eLife.97051.3.sa3
Author response https://doi.org/10.7554/eLife.97051.3.sa4

# Additional files

## Supplementary files

- MDAR checklist
- Supplementary file 1. $^1$H NMR and $^{13}$C NMR Spectra of compounds 1–26.

## Data availability

All data generated or analysed during this study are included in the manuscript and supporting files.

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

# Appendix 1

## Method for synthesis of Group 1–3

### General method for synthesis of Group 1 (chalcone analogues)

A reaction mixture of 3,4,5-trimethoxy acetophenone, benzaldehyde (1.0 equiv.) and 50% NaOH (4.0 equiv.) in EtOH (10 mL) was stirred at RT (24 h). 4 M HCl (16.0 equiv.) (for 50 and 55) was added to the reaction mixture and kept stirring for 20 min. Water was added and reaction mixture was extracted with ethyl acetate and washed with water and brine, successively, and then dried over anhydrous $MgSO_4$. Solvent was removed under reduced pressure and the residue was purified by silica gel column chromatography. (eluent: ethyl acetate:n-hexane)

### (E)–3-(4-Hydroxy-3-methoxyphenyl)–1-(3,4,5-trimethoxyphenyl)prop-2-en-1-one (1)

4-((3,4-Dihydro-2H-pyran-6-yl)oxy)–3-methoxybenzaldehyde (1.00 g, 4.27 mmol) and 3,4,5-trimethoxy acetophenone (0.89 g, 4.27 mmol) and 50% NaOH (1.37 mL, 17.08 mmol) were used. Purification was conducted with eluent (ethyl acetate:n-hexane=1:3) to give compound **1** (0.88 g, 59.8%) as an yellow solid. mp 77–78 °C; $R_f$ 0.69 (ethyl acetate:n-hexane=1:1); HPLC: $R_T$ 5.39 min (purity; 99.9%); ¹H-NMR ($CDCl_3$, 400 MHz) δ 3.94 (s, 3 H), 3.95 (s, 6 H), 3.97 (s, 3 H), 6.97 (d, J=8.0 Hz, 1 H), 7.12 (d, J=1.6 Hz, 1 H), 7.26 (s, 2 H), 7.27 (dd, J=8.0, 1.6 Hz, 1 H), 7.31 (d, J=15.2 Hz, 1 H), 7.75 (d, J=15.2 Hz, 1 H); ¹³C-NMR ($CDCl_3$, 100 MHz) 56.3, 56.7, 61.2, 106.4, 110.7, 115.1, 119.8, 123.2, 127.7, 134.1, 142.6, 145.4, 147.0, 148.5, 153.4, 189.7 ppm.

### (E)–3-(4-(Allyloxy)–3-methoxyphenyl)–1-(3,4,5-trimethoxyphenyl)prop-2-en-1-one (2)

4-(Allyloxy)–3-methoxybenzaldehyde (1.03 g, 5.83 mmol) and 3,4,5-trimethoxy acetophenone (1.22 g, 5.83 mmol) and 50% NaOH (1.87 mL, 23.32 mmol) were used. Purification was conducted with eluent (ethyl acetate:n-hexane=1:3) to give compound **2** (1.99 g, 88.4%) as an pale yellow solid. mp 115–116 °C; $R_f$ 0.62 (ethyl acetate:n-hexane=1:1); HPLC: $R_T$ 11.30 min (purity; 99.9%); ¹H-NMR ($CDCl_3$, 400 MHz) δ 3.94 (s, 3 H), 3.95 (s, 6 H), 3.96 (s, 3 H), 4.68 (ddd, J=5.2, 1.6, 1.2 Hz, 2 H), 5.33 (ddd, J=10.4, 2.4, 1.2 Hz, 1 H), 5.43 (ddd, J=17.2, 2.8, 1.6 Hz, 1 H), 6.04–6.14 (m, 1 H), 6.91 (d, J=8.4 Hz, 1 H), 7.16 (d, J=2.0 Hz, 1 H), 7.23 (dd, J=8.4, 2.0 Hz, 1 H), 7.26 (s, 2 H), 7.32 (d, J=15.2 Hz, 1 H), 7.76 (d, J=15.2 Hz, 1 H); ¹³C-NMR ($CDCl_3$, 100 MHz) 56.3, 56.7, 61.2, 70.3, 106.4, 111.2, 113.2, 118.7, 120.2, 122.9, 128.3, 132.9, 134.1, 142.6, 145.2, 149.8, 150.7, 153.4, 189.7 ppm.

### (E)–3-(4-(But-3-en-1-yloxy)–3-methoxyphenyl)–1-(3,4,5-trimethoxyphenyl)prop-2-en-1-one (3)

4-(But-3-en-1-yloxy)–3-methoxybenzaldehyde (0.31 g, 1.52 mmol) and 3,4,5-trimethoxy acetophe-none (0.32 g, 1.52 mmol) and 50% NaOH (0.48 mL, 1.92 mmol) were used. Purification was conducted with eluent (ethyl acetate:n-hexane=1:3) to give compound **3** (0.34 g, 54.1%) as an pale yellow solid. mp 133–134 °C; $R_f$ 0.69 (ethyl acetate:n-hexane=1:1); HPLC: $R_T$ 13.80 min (purity; 99.9%); ¹H-NMR ($CDCl_3$, 400 MHz) δ 2.50 (dd, J=6.8, 6.8 Hz, 2 H), 3.82 (s, 3 H), 3.83 (s, 3 H), 3.84 (s, 6 H), 4.02 (t, J=6.8 Hz, 2 H), 5.02 (dd, J=11.2, 2.0 Hz, 1 H), 5.08 (dd, J=17.2, 2.0 Hz, 1 H), 5.75–5.86 (m, 1 H), 6.81 (d, J=8.4 Hz, 1 H), 7.07 (d, J=2.0 Hz, 1 H), 7.14 (dd, J=8.4, 2.0 Hz, 1 H), 7.16 (s, 2 H), 7.24 (d, J=16.0 Hz, 1 H), 7.64 (d, J=16.0 Hz, 1 H); ¹³C-NMR (DMSD-$d_6$, 100 MHz) 33.0, 55.9, 56.2, 60.1, 67.5, 106.2, 112.1, 112.8, 117.1, 119.7, 123.4, 127.6, 133.3, 134.7, 141.8, 144.3, 149.1, 150.5, 152.9, 187.9 ppm.

### (E)–3-(4-((E)-But-2-en-1-yloxy)–3-methoxyphenyl)–1-(3,4,5-trimethoxyphenyl)prop-2-en-1-one (4)

(E)–4-(But-2-en-1-yloxy)–3-methoxybenzaldehyde (0.46 g, 2.23 mmol) and 3,4,5-trimethoxy acetophe-none (0.47 g, 2.23 mmol) and 50% NaOH (0.71 mL, 8.92 mmol) were used. Purification was conducted with eluent (ethyl acetate:n-hexane=1:3) to give compound **4** (0.79 g, 87.6%) as an yellow solid. m.p. 130–131 °C; $R_f$ 0.57 (ethyl acetate:n-hexane=1:1); HPLC: $R_T$ 13.69 min (purity; 99.9%); ¹H-NMR ($CDCl_3$, 400 MHz) δ 1.76 (dd, J=6.4, 1.2 Hz, 3 H), 3.94 (s, 6 H), 3.95 (s, 6 H), 4.59 (d, J=6.0 Hz, 2 H), 5.74–5.79 (m, 1 H), 5.85–5.92 (m, 1 H), 6.92 (d, J=8.4 Hz, 1 H), 7.15 (d, J=2.0 Hz, 1 H), 7.22 (dd, J=8.0, 2.0 Hz, 1 H), 7.26 (s, 2 H), 7.32 (d, J=15.6 Hz, 1 H), 7.76 (d, J=15.6 Hz, 1 H);

$^{13}$C-NMR (DMSD-$d_6$, 100 MHz) 17.5, 55.7, 56.2, 60.2, 68.6, 106.2, 111.9, 119.7 123.3, 126.2, 127.5, 130.2, 133.3, 141.8, 144.3, 149.1, 150.2, 152.9, 187.9 ppm.

## (*E*)–3-(3-Methoxy-4-((3-methylbut-2-en-1-yl)oxy)phenyl)–1-(3,4,5-trimethoxy-phenyl) prop-2-en-1-one (5)

3-Methoxy-4-((3-methylbut-2-en-1-yl)oxy)benzaldehyde (0.50 g, 2.27 mmol) and 3,4,5-trimethoxy acetophe-none (0.47 g, 2.27 mmol) and 50% NaOH (0.73 mL, 9.08 mmol) were used. Purification was conducted with eluent (ethyl acetate:*n*-hexane=1:3) to give compound **5** (0.60 g, 61.2%) as an pale yellow solid. mp 115–116 °C; R$_f$ 0.56 (ethyl acetate:n-hexane=1:1); HPLC: R$_T$ 15.71 min (purity; 99.9%); $^1$H-NMR (CDCl$_3$, 400 MHz) δ 1.76 (d, *J*=0.8 Hz, 3 H), 1.79 (d, *J*=0.8 Hz, 3 H), 3.94 (s, 3 H), 3.95 (s, 9 H), 4.65 (d, *J*=6.4 Hz, 2 H), 5.50–5.54 (m, 1 H), 6.91 (d, *J*=8.4 Hz, 1 H), 7.15 (d, *J*=2.0 Hz, 1 H), 7.23 (dd, *J*=8.8, 2.4 Hz, 1 H), 7.26 (s, 2 H), 7.32 (d, *J*=15.6 Hz, 1 H), 7.76 (d, *J*=15.6 Hz, 1 H); $^{13}$C-NMR (CDCl$_3$, 100 MHz) 18.5, 26.1, 56.3, 56.7, 61.2, 66.1, 106.4, 110.7, 110.9, 112.9, 115.1, 119.6, 119.9, 123.0, 123.2, 127.9, 138.5, 145.3, 149.9, 153.4, 187.9 ppm.

## (*E*)–3-(3-Hydroxy-4-methoxyphenyl)–1-(3,4,5-trimethoxyphenyl)prop-2-en-1-one (6)

3-Hydroxy-4-methoxybenzaldehyde (1.0 g, 4.27 mmol) and 3,4,5-trimethoxy acetophenone (0.89 g, 4.27 mmol) and 50% NaOH (1.37 mL, 17.08 mmol) were used. Purification was conducted with eluent (ethyl acetate:*n*-hexane=1:3) to give compound **6** (1.08 g, 73.5%) as an yellow solid. mp 136–137 °C; R$_f$ 0.53 (ethyl acetate:*n*-hexane=1:1); HPLC: R$_T$ 5.50 min (purity; 99.9%); $^1$H-NMR (CDCl$_3$, 400 MHz) δ 3.94 (s, 3 H), 3.95 (s, 9 H), 5.72 (brs, 1 H), 6.89 (d, *J*=8.4 Hz, 1 H), 7.14 (dd, *J*=8.0, 2.0 Hz, 1 H), 7.29 (s, 2 H), 7.31 (d, *J*=2.4 Hz, 1 H), 7.35 (d, *J*=15.6 Hz, 1 H), 7.75 (d, *J*=15.6 Hz, 1 H); $^{13}$C-NMR (CDCl$_3$, 100 MHz) 51.1, 56.3, 56.6, 61.2, 106.2, 110.8, 112.9, 120.1, 123.2, 128.8, 133.9, 142.6, 144.9, 146.1, 149.1, 153.4, 189.3 ppm.

## (*E*)–3-(3-(Allyloxy)–4-methoxyphenyl)–1-(3,4,5-trimethoxyphenyl)prop-2-en-1-one (7)

3-(Allyloxy)–4-methoxybenzaldehyde (0.77 g, 4.00 mmol) and 3,4,5-trimethoxy acetophenone (0.84 g, 4.00 mmol) and 50% NaOH (1.28 mL, 16.00 mmol) were used. Purification was conducted with eluent (ethyl acetate:*n*-hexane=1:3) to give compound **7** (0.85 g, 55.3%) as an yellow solid. mp 89–90 °C; R$_f$ 0.67 (ethyl acetate:*n*-hexane=1:1); HPLC: R$_T$ 10.85 min (purity; 99.9%); $^1$H-NMR (CDCl$_3$, 400 MHz) δ 3.93 (s, 3 H), 3.94 (s, 3 H), 3.95 (s, 6 H), 4.68 (dt, *J*=5.6, 1.6 Hz, 2 H), 5.33 (ddt, *J*=10.8, 2.8, 1.6 Hz, 1 H), 5.45 (ddt, *J*=16.8, 2.8, 1.6 Hz, 1 H), 6.07–6.16 (m, 1 H), 6.92 (d, *J*=8.4 Hz, 1 H), 7.18 (d, *J*=2.0 Hz, 1 H), 7.25 (dd, *J*=8.4, 2.0 Hz, 1 H), 7.26 (s, 2 H), 7.30 (d, *J*=15.6 Hz, 1 H), 7.75 (d, *J*=15.6 Hz, 1 H); $^{13}$C-NMR (DMSO-$d_6$, 100 MHz) 55.7, 56.2, 60.2, 69.1, 106.2, 111.9, 113.2, 117.8, 119.7, 123.7, 127.5, 133.3, 133.8, 141.8, 144.3, 147.7, 151.5, 152.9, 187.9 ppm.

## (*E*)–3-(3-(But-3-en-1-yloxy)–4-methoxyphenyl)–1-(3,4,5-trimethoxyphenyl) prop-2-en-1-one (8)

3-(But-3-en-1-yloxy)–4-methoxybenzaldehyde (0.10 g, 0.48 mmol) and 3,4,5-trimethoxy aceto-phenone (0.10 g, 0.48 mmol) and 50% NaOH (0.15 mL, 1.92 mmol) were used. Purification was conducted with eluent (ethyl acetate:*n*-hexane=1:3) to give compound **8** (0.10 g, 54.2%) as an yellow solid. m.p. 104–105 °C; R$_f$ 0.67 (ethyl acetate:*n*-hexane=1:1); HPLC: R$_T$ 13.13 min (purity; 99.9%); $^1$H-NMR (CDCl$_3$, 400 MHz) δ 2.59–2.67 (m, 2 H), 3.92 (s, 3 H), 3.94 (s, 3 H), 3.95 (s, 6 H), 4.13 (t, *J*=6.8 Hz, 2 H), 5.14 (ddt, *J*=10.4, 3.2, 1.2 Hz, 1 H), 5.21 (ddt, *J*=17.2, 3.2, 1.2 Hz, 1 H), 5.89–5.99 (m, 1 H), 6.92 (d, *J*=8.8 Hz, 1 H), 7.18 (d, *J*=1.6 Hz, 1 H), 7.25 (dd, *J*=8.8, 1.6 Hz, 1 H), 7.29 (s, 2 H), 7.30 (d, *J*=15.6 Hz, 1 H), 7.75 (d, *J*=15.6 Hz, 1 H); $^{13}$C-NMR (DMSD-$d_6$, 100 MHz) 33.1, 55.7, 56.2, 60.2, 67.7, 106.2, 111.9, 113.1, 116.9, 119.7, 123.5, 127.6, 133.3, 134.9, 141.8, 144.3, 148.1, 151.5, 152.9, 187.9 ppm.

## (*E*)–3-(3-((E)-But-2-en-1-yloxy)–4-methoxyphenyl)–1-(3,4,5-trimethoxyphenyl) prop-2-en-1-one (9)

(*E*)–3-(But-2-en-1-yloxy)–4-methoxybenzaldehyde (0.50 g, 2.42 mmol) and 3,4,5-trimethoxy aceto-phenone (0.51 g, 2.42 mmol) and 50% NaOH (0.76 mL, 12.10 mmol) were used. Purification was conducted with eluent (ethyl acetate:*n*-hexane=1:3) to give compound **9** (0.46 g, 47.8%) as an yellow

solid. m.p. 92–93 °C; $R_f$ 0.57 (ethyl acetate:*n*-hexane=1:1); HPLC: $R_T$ 13.13 min (purity; 99.9%); [1]H-NMR (CDCl$_3$, 400 MHz) δ 1.77 (dd, *J*=6.8, 1.6 Hz, 3 H), 3.93 (s, 3 H), 3.94 (s, 3 H), 3.95 (s, 6 H), 4.59 (d, *J*=6.4 Hz, 2 H), 5.71–5.83 (m, 1 H), 5.86–5.94 (m, 1 H), 6.91 (d, *J*=8.4 Hz, 1 H), 7.17 (d, *J*=2.0 Hz, 1 H), 7.24 (dd, *J*=8.4, 2.0 Hz, 1 H), 7.26 (s, 2 H), 7.30 (d, *J*=15.6 Hz, 1 H), 7.75 (d, *J*=15.6 Hz, 1 H); [13]C-NMR (DMSD-$d_6$, 100 MHz) 17.5, 55.6, 56.2, 60.2, 68.8, 106.2, 111.8, 112.9, 119.6, 123.5, 126.4, 127.4, 130.1, 133.3, 141.8, 144.3, 147.8, 151.5, 152.9, 187.9 ppm.

### (*E*)–3-(4-Methoxy-3-((3-methylbut-2-en-1-yl)oxy)phenyl)–1-(3,4,5-trimethoxy-phenyl) prop-2-en-1-one (10)

4-Methoxy-3-((3-methylbut-2-en-1-yl)oxy)benzaldehyde (0.50 g, 2.27 mmol) and 3,4,5-trimethoxy acetophenone (0.48 g, 2.27 mmol) and 50% NaOH (0.73 mL, 9.08 mmol) were used. Purification was conducted with eluent (ethyl acetate:*n*-hexane=1:3) to give compound **10** (0.64 g, 47.8%) as an yellow solid. mp 116–117 °C; $R_f$ 0.62 (ethyl acetate:*n*-hexane=1:1); HPLC: $R_T$ 15.05 min (purity; 99.9%); [1]H-NMR (CDCl$_3$, 400 MHz) δ 1.78 (s, 3 H), 1.79 (s, 3 H), 3.92 (s, 3 H), 3.94 (s, 3 H), 3.95 (s, 6 H), 4.65 (d, *J*=6.8 Hz, 2 H), 5.52–5.56 (m, 1 H), 6.90 (d, *J*=8.4 Hz, 1 H), 7.17 (d, *J*=1.6 Hz, 1 H), 7.25 (dd, *J*=8.4, 1.6 Hz, 1 H), 7.26 (s, 2 H), 7.31 (d, *J*=15.6 Hz, 1 H), 7.75 (d, *J*=15.6 Hz, 1 H); [13]C-NMR (DMSD-$d_6$, 100 MHz) 18.0, 25.4, 55.6, 56.2, 60.2, 65.1, 106.1, 111.7, 112.9, 119.6, 123.5, 127.4, 133.3, 137.3, 141.8, 144.4, 147.9, 151.5, 152.9, 187.9 ppm.

## General method for synthesis of Group 2 (pyrazoline analogues)

A reaction mixture of chalcone analogue and hydrazine H$_2$O(65%) (4.0 equiv.) in AcOH (10 mL) was refluxed (2 h) and then cooled to room temperature. There action mixture was poured into ice and kept overnight at room temperature. Solid formed was filtered and washed with H$_2$O. Solid was dried and purified by silica gel column chromatography (eluent: methanol: chloroform).

### 1-(5-(4-Hydroxy-3-methoxyphenyl)–3-(3,4,5-trimethoxyphenyl)–4,5-dihydro-1*H*-pyrazol-1-yl)propan-1-one (11)

Compound **1** (0.10 g, 0.29 mmol) and hydrazine H$_2$O (65%) (0.078 mL, 1.16 mmol) were used. Purification was conducted with eluent (methanol:chloroform = 1:100) to give compound **11** (0.078 g, 65.0%) as an yellow solid. mp 178–179 °C; $R_f$ 0.48 (methanol: chloroform = 1:10); HPLC: $R_T$ 3.57 min (purity; 95.5%); [1]H-NMR (CDCl$_3$, 400 MHz) δ 2.42 (s, 3 H), 3.15 (dd, *J*=17.6, 4.4 Hz, 1 H), 3.71 (dd, *J*=17.6, 12.0 Hz, 1 H), 3.87 (s, 3 H), 3.89 (s, 3 H), 3.91 (s, 6 H), 5.51 (dd, *J*=11.6, 4.4 Hz, 1 H), 5.55 (s, 1 H), 6.72 (dd, *J*=8.4, 2.0 Hz, 1 H), 6.76 (d, *J*=1.6 Hz, 1 H), 6.85 (d, *J*=8.0 Hz, 1 H), 6.96 (s, 2 H); [13]C-NMR (DMSO-d$_6$, 100 MHz) 21.7, 42.3, 55.6, 55.9, 59.3, 60.1, 104.1, 110.0, 115.5, 117.3, 126.7, 133.4, 139.4, 145.7, 147.5, 153.0, 154.2, 167.2 ppm.

### 1-(5-(4-(Allyloxy)–3-methoxyphenyl)–3-(3,4,5-trimethoxyphenyl)–4,5-dihy-dro-1*H*-pyrazol-1-yl)ethanone (12)

Compound **2** (0.50 g, 1.30 mmol) and hydrazine H$_2$O (65%) (0.35 mL, 5.20 mmol) were used. Purification was conducted with eluent (methanol:chloroform = 1 : 80) to give compound **12** (0.33 g, 57.4%) as an yellow syrup. $R_f$ 0.58 (methanol:chloroform = 1:9); HPLC: $R_T$ 7.57 min (purity; 96.0%); [1]H-NMR (CDCl$_3$, 400 MHz) δ 2.73 (s, 3 H), 3.44 (dd, *J*=17.6, 4.4 Hz, 1 H), 4.01 (dd, *J*=17.6, 11.6 Hz, 1 H), 4.15 (s, 3 H), 4.19 (s, 3 H), 4.21 (s, 6 H), 4.87 (d, *J*=5.2 Hz, 2 H), 5.56 (dd, *J*=10.4, 0.8 Hz, 1 H), 5.67 (dd, *J*=17.2, 1.6 Hz, 1 H), 5.84 (dd, *J*=11.6, 4.4 Hz, 1 H), 6.31–6.38 (m, 1 H), 7.03 (dd, *J*=8.4, 2.0 Hz, 1 H), 7.07 (d, *J*=2.0 Hz, 1 H), 7.11 (d, *J*=8.4 Hz, 1 H), 7.26 (s, 2 H); [13]C-NMR (DMSO-d$_6$, 100 MHz) 21.7, 42.3, 55.6, 56.0, 59.2, 60.1, 69.0, 104.1, 109.9, 113.8, 116.9, 117.4, 126.6, 133.9, 135.3, 139.4, 146.7, 149.1, 153.0, 154.2, 167.3 ppm.

### 1-(5-(4-(But-3-en-1-yloxy)–3-methoxyphenyl)–3-(3,4,5-trimethoxyphenyl)–4,5-dihydro-1*H*-pyrazol-1-yl)ethanone (13)

Compound **3** (0.10 g, 0.25 mmol) and hydrazine H$_2$O (65%) (0.07 mL, 1.00 mmol) were used. Purification was conducted with eluent (methanol:chloroform = 1:100) to give compound **13** (0.066 g, 60.0%) as an yellow lsyrup. $R_f$ 0.34 (methanol:chloroform = 1:10); HPLC: $R_T$ 9.02 min (purity; 99.9%); [1]H-NMR (CDCl$_3$, 400 MHz) δ 2.31 (s, 3 H), 2.45 (dt, *J*=6.8, 1.6 Hz, 2 H), 3.69 (s, 3 H), 3.73 (s, 3 H), 3.82 (s, 6 H), 3.96 (t, *J*=6.8 Hz, 2 H), 5.05–5.08 (m, 1 H), 5.12–5.18 (m, 1 H), 5.49 (dd, *J*=11.6, 4.4 Hz, 1 H), 5.82–5.92 (m, 1 H), 6.62 (dd, *J*=8.4, 2.0 Hz, 1 H), 6.81 (d, *J*=2.0 Hz, 1 H), 6.89 (d, *J*=8.4 Hz, 1 H), 7.05 (s, 2 H); [13]C-NMR (DMSO-d$_6$, 100 MHz) 21.7, 33.1, 42.3, 55.6, 56.0, 59.2, 60.1,

67.7, 104.1, 110.0, 113.6, 116.9, 117.0, 126.6, 134.9, 135.3, 139.4, 147.1, 149.1, 152.9, 154.2, 167.3 ppm.

### (*E*)–1-(5-(4-(But-2-en-1-yloxy)–3-methoxyphenyl)–3-(3,4,5-trimethoxyphenyl)–4,5-dihydro-1*H*-pyrazol-1-yl)ethanone (14)

Compound **4** (0.50 g, 1.25 mmol) and hydrazine $H_2O$ (65%) (0.34 mL, 5.00 mmol) were used. Purification was conducted with eluent (methanol:chloroform = 1:80) to give compound **14** (0.29 g, 50.8%) as an yellow syrup. $R_f$ 0.68 (methanol:chloroform = 1:9); HPLC: $R_T$ 9.02 min (purity; 98.3%); $^{1}$H-NMR (CDCl$_3$, 400 MHz) δ 1.72 (dd, *J*=6.0, 1.2 Hz, 3 H), 2.43 (s, 3 H), 3.14 (dd, *J*=17.6, 4.8 Hz, 1 H), 3.71 (dd, *J*=16.8, 12.0 Hz, 1 H), 3.84 (s, 3 H), 3.89 (s, 3 H), 3.91 (s, 6 H), 4.48 (d, *J*=6.0 Hz, 2 H), 5.54 (dd, *J*=11.6, 4.4 Hz, 1 H), 5.69–5.85 (m, 2 H), 6.73(dd, *J*=8.4, 2.0 Hz, 1 H), 6.76 (d, *J*=2.0 Hz, 1 H), 6.81 (d, *J*=8.4 Hz, 1 H), 6.96 (s, 2 H); $^{13}$C-NMR (DMSO-d$_6$, 100 MHz) 17.5, 21.7, 42.3, 55.5, 55.9, 59.2, 60.1, 68.7, 104.1, 109.8, 113.5, 116.9, 126.5, 126.6, 129.5, 135.1, 139.4, 146.9, 149.1, 152.9, 154.2, 167.3 ppm.

### 1-(5-(3-Hydroxy-4-methoxyphenyl)–3-(3,4,5-trimethoxyphenyl)–4,5-dihydro-1*H*-pyrazol-1-yl)ethanone (15)

Compound **6** (0.10 g, 0.29 mmol) and hydrazine $H_2O$ (65%) (0.078 mL, 1.16 mmol) were used. Purification was conducted with eluent (methanol:chloroform = 1:80) to give compound **15** (0.096 g, 82.6%) as an white solid. m.p.182–183 °C; $R_f$ 0.45 (methanol:chloroform = 1:10); HPLC: $R_T$ 3.77 min (purity; 98.0%); $^{1}$H-NMR (CDCl$_3$, 400 MHz) δ 2.42 (s, 3 H), 3.12 (dd, *J*=17.6, 4.4 Hz, 1 H), 3.70 (dd, *J*=17.6, 11.6 Hz, 1 H), 3.86 (s, 3 H), 3.89 (s, 3 H), 3.91 (s, 6 H), 5.51 (dd, *J*=11.6, 4.4 Hz, 1 H), 5.59 (s, 1 H), 6.75–6.80 (m, 3 H), 6.96 (s, 2 H); $^{13}$C-NMR (DMSO-d$_6$, 100 MHz) 21.7, 42.2, 55.7, 55.9, 59.0, 60.1, 104.1, 110.0, 112.4, 112.5, 126.6, 135.2, 139.4, 146.7, 146.8, 153.0, 154.2, 167.1 ppm.

### 1-(5-(3-(Allyloxy)–4-methoxyphenyl)–3-(3,4,5-trimethoxyphenyl)–4,5-dihydro-1*H*-pyrazol-1-yl)ethanone (16)

Compound **7** (0.42 g, 1.10 mmol) and hydrazine $H_2O$ (65%) (0.29 mL, 4.40 mmol) were used. Purification was conducted with eluent (methanol:chloroform = 1:80) to give compound **16** (0.33 g, 68.4%) as an yellow syrup. m.p. °C; $R_f$ 0.58 (methanol:chloroform = 1:10); HPLC: RT 7.32 min (purity; 95.3%); $^{1}$H-NMR (CDCl$_3$, 400 MHz) δ 2.42 (s, 3 H), 3.13 (dd, *J*=17.2, 4.4 Hz, 1 H), 3.70 (dd, *J*=17.6, 12.0 Hz, 1 H), 3.83 (s, 3 H), 3.89 (s, 3 H), 3.92(s, 6 H), 4.58 (d, *J*=5.6 Hz, 2 H), 5.25 (ddt, *J*=10.8, 2.4, 1.6 Hz, 1 H), 5.36 (ddt, *J*=17.2, 3.2, 1.6 Hz, 1 H), 5.52 (dd, *J*=11.6, 4.4 Hz, 1 H), 6.00–6.11 (m, 1 H), 6.77–6.82 (m, 3 H), 6.96 (s, 2 H); $^{13}$C-NMR (CDCl$_3$, 100 MHz) 22.2, 42.7, 56.2, 56.5, 59.9, 61.2, 70.3, 104.2, 111.6, 112.1, 118.3, 127.1, 133.6, 134.7, 140.5, 148.5, 149.2, 153.6, 153.9, 168.9 ppm.

### (*E*)–1-(5-(3-(But-2-en-1-yloxy)–4-methoxyphenyl)–3-(3,4,5-trimethoxyphenyl)–4,5-dihydro-1*H*-pyrazol-1-yl)ethanone (17)

Compound **9** (0.44 g, 1.10 mmol) and hydrazine $H_2O$ (65%) (0.29 mL, 4.40 mmol) were used. Purification was conducted with eluent (methanol:chloroform = 1:80) to give compound **17** (0.28 g, 56.6%) as an yellow syrup. $R_f$ 0.68 (methanol:chloroform = 1:9); HPLC: $R_T$ 8.29 min (purity; 99.8%); $^{1}$H-NMR (CDCl$_3$, 400 MHz) δ 1.69 (dd, *J*=6.0, 1.2 Hz, 3 H), 2.43 (s, 3 H), 3.13 (dd, *J*=17.6, 4.4 Hz, 1 H), 3.71 (dd, *J*=17.6, 12.0 Hz, 1 H), 3.82 (s, 3 H), 3.89 (s, 3 H), 3.91 (s, 6 H), 4.48 (d, *J*=6.0 Hz, 2 H), 5.53 (dd, *J*=12.0, 4.4 Hz, 1 H), 5.67–5.83 (m, 2 H), 6.76–6.82 (m, 3 H), 6.96 (s, 2 H); $^{13}$C-NMR (CDCl$_3$, 100 MHz) 18.0, 22.2, 42.7, 56.2, 56.5, 60.0, 61.2, 69.9, 104.2, 111.2, 111.9, 118.1, 126.3, 128.6, 131.1, 134.6, 140.5, 148.6, 149.2, 153.6, 153.9, 168.9 ppm.

### General method for synthesis of Group 3 (pyrazoline analogues)

A reaction mixture of chalcone analogue and hydrazine $H_2O$ (65%) (4.0 equiv.) in propionic acid (10 mL) was refluxed (2 h) and then cooled to room temperature. The reaction mixture was poured into ice and kept overnight at room temperature. Solid formed was filtered and washed with $H_2O$. Solid was dried and purified by silica gel column chromatography (eluent: methanol:chloroform).

### 1-(5-(4-Hydroxy-3-methoxyphenyl)–3-(3,4,5-trimethoxyphenyl)–4,5-dihydro-1*H*-pyrazol-1-yl)ethanone (18)

Compound **1** (0.10 g, 0.48 mmol) and hydrazine $H_2O$ (65%) (0.13 mL, 1.92 mmol) were used. Purification was conducted with eluent (methanol:chloroform = 1:80) to give compound **18** (0.088 g, 75.5%) as an pale yellow solid. mp 180–181 °C; $R_f$ 0.44 (methanol:chloroform = 1:10); HPLC: $R_T$ 4.75 min (purity; 99.9%); ¹H-NMR (CDCl₃, 400 MHz) δ 1.20 (t, *J*=7.2 Hz, 3 H), 2.81 (q, *J*=7.2 Hz, 2 H), 3.13 (dd, *J*=17.6, 4.4 Hz, 1 H), 3.70 (dd, *J*=17.6, 11.6 Hz, 1 H), 3.87 (s, 3 H), 3.89 (s, 3 H), 3.91 (s, 6 H), 5.50 (dd, *J*=12.0, 4.8 Hz, 1 H), 5.54 (s, 1 H), 6.72 (dd, *J*=8.4, 2.0 Hz, 1 H), 6.75 (d, *J*=1.6 Hz, 1 H), 6.85 (d, *J*=8.0 Hz, 1 H), 6.96 (s, 2 H); ¹³C-NMR (DMSO-d₆, 100 MHz) 9.0, 26.8, 42.1, 55.6, 55.9, 59.3, 60.1, 104.0, 109.9, 115.5, 117.3, 126.8, 133.6, 139.4, 145.7, 147.5, 153.0, 154.0, 170.6 ppm.

### 1-(5-(4-(Allyloxy)–3-methoxyphenyl)–3-(3,4,5-trimethoxyphenyl)–4,5-dihydro-1*H*-pyrazol-1-yl)propan-1-one (19)

Compound **2** (0.20 g, 0.26 mmol) and hydrazine $H_2O$ (65%) (0.14 mL, 1.04 mmol) were used. Purification was conducted with eluent (methanol:chloroform = 1:80) to give compound **19** (0.094 g, 79.7%) as an yellow syrup. $R_f$ 0.37(methanol:chloroform = 1:10); HPLC: $R_T$ 9.37 min (purity; 97.6%); ¹H-NMR (CDCl₃, 400 MHz) δ 1.19 (t, *J*=7.6 Hz, 3 H), 2.81 (q, *J*=7.6 Hz, 2 H), 3.11 (dd, *J*=17.6, 4.8 Hz, 1 H), 3.68 (dd, *J*=17.6, 12.0 Hz, 1 H), 3.83 (s, 3 H), 3.88 (s, 3 H), 3.89 (s, 6 H), 4.55 (dt, *J*=5.2, 1.6 Hz, 2 H), 5.24 (dq, *J*=10.4, 1.6 Hz, 1 H), 5.35 (dq, *J*=16.0, 1.6 Hz, 1 H), 5.51 (dd, *J*=11.2, 4.4 Hz, 1 H), 5.99–6.07 (m, 1 H), 6.71 (dd, *J*=8.0, 2.0 Hz, 1 H), 6.73 (d, *J*=2.0 Hz, 1 H), 6.89 (d, *J*=8.4 Hz, 1 H), 6.94 (s, 2 H); ¹³C-NMR (DMSO-d₆, 100 MHz) 9.0, 26.8, 42.1, 55.5, 55.9, 59.3, 60.1, 69.0, 104.1, 109.8, 113.8, 116.9, 117.4, 126.7, 133.9, 135.4, 139.4, 146.7, 149.1, 153.0, 154.0, 170.6 ppm.

### 1-(5-(4-(But-3-en-1-yloxy)–3-methoxyphenyl)–3-(3,4,5-trimethoxyphenyl)–4,5-dihydro-1*H*-pyrazol-1-yl)propan-1-one (20)

Compound **3** (0.10 g, 0.25 mmol) and hydrazine $H_2O$ (65%) (0.07 mL, 1.00 mmol) were used. Purification was conducted with eluent (methanol:chloroform = 1:80) to give compound **201** (0.063 g, 53.4%) as an yellow syrup. $R_f$ 0.71 (methanol:chloroform = 1:10); HPLC: $R_T$ 4.36 min (purity; 96.7%); ¹H-NMR (DMSO-d₆, 400 MHz) δ 1.07 (t, *J*=7.6 Hz, 3 H), 2.45 (qt, *J*=6.8, 1.2 Hz, 2 H), 2.51~2.79 (m, 2 H), 3.19 (dd, *J*=17.6, 4.8 Hz, 1 H), 3.69 (s, 3 H), 3.73 (s, 3 H), 3.82 (s, 6 H), 3.79 (dd, *J*=17.6, 6.8 Hz, 1 H), 3.96 (t, *J*=6.8 Hz, 2 H), 5.08 (dq, *J*=10.0, 1.2 Hz, 1 H), 5.15 (dq, *J*=17.2, 1.6 Hz, 1 H), 5.48 (dd, *J*=12.0, 4.0 Hz, 1 H), 5.82–5.92 (m, 1 H), 6.62 (dd, *J*=8.4, 2.0 Hz, 1 H), 6.80 (d, *J*=2.0 Hz, 1 H), 6.90 (d, *J*=8.4 Hz, 1 H), 7.04 (s, 2 H); ¹³C-NMR (DMSO-d₆, 100 MHz) 9.0, 26.8, 33.1, 42.1, 55.6, 55.9, 59.3, 60.1, 67.7, 104.1, 109.9, 116.9, 117.0, 126.7, 134.9, 135.4, 139.4, 147.1, 149.1, 152.9, 154.0, 170.6 \ ppm.

### (*E*)–1-(5-(4-(But-2-en-1-yloxy)–3-methoxyphenyl)–3-(3,4,5-trimethoxyphenyl)–4,5-dihydro-1*H*-pyrazol-1-yl)propan-1-one (21)

Compound **4** (0.10 g, 0.25 mmol) and hydrazine $H_2O$ (65%) (0.07 mL, 1.00 mmol) were used. Purification was conducted with eluent (methanol:chloroform = 1:100) to give compound **21** (0.042 g, 35.2%) as an yellow semisolid. $R_f$ 0.84 (methanol:chloroform = 1:10); HPLC: $R_T$ 4.54 min (purity; 95.3%); ¹H-NMR (DMSO-d₆, 400 MHz) δ 1.07 (t, *J*=7.6 Hz, 3 H), 1.68 (dd, *J*=6.8, 1.6 Hz, 3 H), 2.68–2.81 (m, 2 H), 3.19 (dd, *J*=18.0, 4.4 Hz, 1 H), 3.67 (dd, *J*=17.6, 12.0 Hz, 1 H), 3.69 (s, 3 H), 3.72 (s, 3 H), 3.76 (dd, *J*=17.2, 6.0 Hz, 1 H), 3.80 (s, 6 H), 4.41 (d, *J*=6.0 Hz, 2 H), 5.48 (dd, *J*=11.6, 4.0 Hz, 1 H), 5.63–5.70 (m, 1 H), 5.76–5.83 (m, 1 H), 6.60 (dd, *J*=8.0, 2.4 Hz, 1 H), 6.78 (d, *J*=2.0 Hz, 1 H), 6.88 (d, *J*=8.0 Hz, 1 H), 7.04 (s, 2 H); ¹³C-NMR (DMSO-d₆, 100 MHz) 9.0, 17.5, 26.8, 42.1, 55.5, 56.0, 59.3, 60.1, 68.7, 104.1, 109.7, 113.5, 116.8, 126.6, 126.7, 135.2, 139.4, 146.8, 149.1, 153.0, 154.0, 170.6 ppm.

### 1-(5-(3-Methoxy-4-((3-methylbut-2-en-1-yl)oxy)phenyl)–3-(3,4,5-trimethoxyphenyl)–4,5-dihydro-1*H*-pyrazol-1-yl)propan-1-one (22)

Compound **5** (0.10 g, 0.24 mmol) and hydrazine $H_2O$ (65%) (0.07 mL, 1.00 mmol) were used. Purification was conducted with eluent (methanol:chloroform = 1:80) to give compound **22** (0.050 g, 43.0%) as an yellow semisolid. $R_f$ 0.39 (methanol:chloroform = 1:10); HPLC: $R_T$ 3.35 min (purity; 99.8%); ¹H-NMR (DMSO-d₆, 400 MHz) δ 1.07 (t, *J*=6.8 Hz, 3 H), 1.67 (s, 3 H), 1.73 (s, 3 H), 2.69–2.79 (m, 2 H), 3.18 (dd, *J*=18.0, 4.4 Hz, 1 H), 3.69 (s, 3 H), 3.73 (s, 3 H), 3.68 (dd, *J*=17.6, 11.6 Hz, 1 H), 3.82 (s, 6 H), 4.46 (d, *J*=6.8 Hz, 2 H), 5.38–5.42 (m, 1 H), 5.48 (dd, *J*=11.6, 4.4 Hz, 1 H), 6.61 (dd, *J*=8.4,

2.0 Hz, 1 H), 6.78 (d, $J$=2.0 Hz, 1 H), 6.88 (d, $J$=8.0 Hz, 1 H), 7.04 (s, 2 H); $^{13}$C-NMR (DMSO-$d_6$, 100 MHz) 9.5, 18.4, 25.9, 27.2, 42.5, 55.9, 56.4, 59.7, 60.6, 65.4, 104.5, 110.1, 113.9, 117.3, 120.6, 127.1, 135.6, 137.3, 139.8, 147.4, 149.5, 153.4, 154.2, 171.0 ppm.

### 1-(5-(3-Hydroxy-4-methoxyphenyl)–3-(3,4,5-trimethoxyphenyl)–4,5-dihydro-1$H$-pyrazol-1-yl)propan-1-one (23)

Compound **6** (0.10 g, 0.29 mmol) and hydrazine H$_2$O (65%) (0.078 mL, 1.16 mmol) were used. Purification was conducted with eluent (methanol:chloroform = 1:80) to give compound **23** (0.059 g, 48.8%) as an white solid. mp 237–238 °C; R$_f$ 0.39 (methanol:chloroform = 1:10); HPLC: R$_T$ 4.69 min (purity; 99.9%); $^1$H-NMR (CDCl$_3$, 400 MHz) δ 1.11 (t, $J$=7.6 Hz, 3 H), 2.74 (q, $J$=7.6 Hz, 2 H), 3.03 (dd, $J$=17.6, 4.4 Hz, 1 H), 3.60 (dd, $J$=17.6, 11.6 Hz, 1 H), 3.77 (s, 3 H), 3.80 (s, 3 H), 3.83 (s, 6 H), 5.40 (dd, $J$=11.6, 4.4 Hz, 1 H), 6.60 (s, 1 H), 6.65 (dd, $J$=8.0, 2.4 Hz, 1 H), 6.68 (d, $J$=2.0 Hz, 1 H), 6.71 (d, $J$=8.0 Hz, 1 H), 6.88 (s, 2 H); $^{13}$C-NMR (DMSO-$d_6$, 100 MHz) 8.9, 26.7, 42.0, 55.7, 56.0, 59.1, 60.1, 104.0, 112.4, 112.5, 116.1, 126.7, 135.3, 139.4, 146.7, 146.8, 153.0, 154.0, 170.4 ppm.

### 1-(5-(3-(Allyloxy)–4-methoxyphenyl)–3-(3,4,5-trimethoxyphenyl)–4,5-dihydro-1$H$-pyrazol-1-yl)propan-1-one (24)

Compound **7** (0.30 g, 0.78 mmol) and hydrazine H$_2$O (65%) (0.21 mL, 3.12 mmol) were used. Purification was conducted with eluent (methanol:chloroform = 1:100) to give compound **24** (0.13 g, 37.1%) as an yellow semisolid. R$_f$ 0.41 (methanol:chloroform = 1:10); HPLC: R$_T$ 3.51 min (purity; 97.5%); $^1$H-NMR (DMSO-$d_6$, 400 MHz) δ 1.06 (t, $J$=7.6 Hz, 3 H), 2.82 (dq, $J$=18.4, 7.2 Hz, 2 H), 3.18 (dd, $J$=17.6, 4.4 Hz, 1 H), 3.70 (s, 3 H), 3.73 (s, 3 H), 3.79 (dd, $J$=18.0, 6.4 Hz, 1 H), 3.82 (s, 6 H), 4.50 (dd, $J$=5.2, 1.2 Hz, 2 H), 5.22 (dq, $J$=10.8, 1.6 Hz, 1 H), 5.36 (dq, $J$=15.6, 1.6 Hz, 1 H), 5.47 (dd, $J$=11.6, 4.4 Hz, 1 H), 5.97–6.07 (m, 1 H), 6.67 (dd, $J$=8.8, 1.6 Hz, 1 H), 6.79 (d, $J$=2.0 Hz, 1 H), 6.90 (d, $J$=8.0 Hz, 1 H), 7.04 (s, 2 H); $^{13}$C-NMR (DMSO-$d_6$, 100 MHz) 9.0, 26.8, 42.1, 55.7, 56.0, 59.2, 60.1, 69.0, 104.1, 111.3, 112.3, 117.5, 117.6, 126.7, 133.8, 135.0, 139.4, 147.7, 148.3, 153.0, 154.0, 170.6 ppm.

### (E)–1-(5-(3-(But-2-en-1-yloxy)–4-methoxyphenyl)–3-(3,4,5-trimethoxyphenyl)–4,5-dihydro-1$H$-pyrazol-1-yl)propan-1-one (25)

Compound **9** (1.50 g, 3.76 mmol) and hydrazine H$_2$O (65%) (1.01 mL, 15.04 mmol) were used. Purification was conducted with eluent (methanol:chloroform = 1:80) to give compound **6** (0.078 g, 4.9%) as an yellow semisolid. R$_f$ 0.39 (methanol:chloroform = 1:9); HPLC: R$_T$ 4.15 min (purity; 97.5%); $^1$H-NMR (DMSO-$d_6$, 400 MHz) δ 1.07 (t, $J$=7.6 Hz, 3 H), 1.65 (dd, $J$=6.4, 1.2 Hz, 3 H), 2.75 (dq, $J$=16.0, 8.0 Hz, 2 H), 3.19 (dd, $J$=18.0, 4.4 Hz, 1 H), 3.69 (s, 3 H), 3.72 (s, 3 H), 3.78 (dd, $J$=18.0, 6.0 Hz, 1 H), 3.82 (s, 6 H), 4.41 (d, $J$=6.0 Hz, 2 H), 5.47 (dd, $J$=11.6, 4.4 Hz, 1 H), 5.60–5.68 (m, 1 H), 5.73–5.82 (m, 1 H), 6.66 (dd, $J$=8.0, 1.6 Hz, 1 H), 6.78 (d, $J$=2.0 Hz, 1 H), 6.88 (d, $J$=8.4 Hz, 1 H), 7.04 (s, 2 H); $^{13}$C-NMR (DMSO-$d_6$, 100 MHz) 9.0, 17.4, 26.8, 42.0, 55.6, 56.0, 59.2, 60.1, 68.6, 104.1, 110.9, 112.1, 117.3, 126.5, 126.7, 129.8, 134.9, 139.4, 147.8, 148.2, 153.0, 154.0, 170.5 ppm.

### 1-(5-(4-Methoxy-3-((3-methylbut-2-en-1-yl)oxy)phenyl)–3-(3,4,5-trimethoxyphenyl)–4,5-dihydro-1$H$-pyrazol-1-yl)propan-1-one (26)

Compound **10** (0.20 g, 0.48 mmol) and hydrazine H$_2$O (65%) (0.13 mL, 1.92 mmol) were used. Purification was conducted with eluent (methanol:chloroform = 1:80) to give compound **26** (0.049 g, 21.3%) as an yellow semi-solid. R$_f$ 0.69 (methanol:chloroform = 1:9); HPLC: R$_T$ 4.62 min (purity; 95.4%); $^1$H-NMR (DMSO-$d_6$, 400 MHz) δ 1.07 (t, $J$=7.6 Hz, 3 H), 1.65 (s, 3 H), 1.67 (s, 3 H), 2.74 (dq, $J$=16.0, 8.0 Hz, 2 H), 3.17 (dd, $J$=18.0, 4.4 Hz, 1 H), 3.69 (s, 3 H), 3.71 (s, 3 H), 3.75 (dd, $J$=18.0, 8.8 Hz, 1 H), 3.80 (s, 6 H), 4.48 (d, $J$=6.8 Hz, 2 H), 5.34 (dd, $J$=6.4, 5.2 Hz, 1 H), 5.47 (dd, $J$=11.6, 4.0 Hz, 1 H), 6.66 (dd, $J$=8.4, 2.0 Hz, 1 H), 6.73 (d, $J$=2.0 Hz, 1 H), 6.88 (d, $J$=8.4 Hz, 1 H), 7.04 (s, 2 H); $^{13}$C-NMR (DMSO-$d_6$, 100 MHz) 9.0, 17.9, 25.3, 26.8, 42.0, 55.6, 56.0, 59.3, 60.1, 64.9, 104.1, 110.7, 112.0, 117.3, 120.1, 126.7, 135.0, 136.8, 139.4, 147.9, 148.3, 153.0, 154.0, 170.5 ppm.

