## [Editor Report · eLife assessment]

This **valuable** study characterized a new set of small molecules targeting the interaction between ELF3-MED23, with one of the reported compounds representing a promising novel therapeutic strategy, The evidence supporting the conclusions is **convincing**. This article will be of interest to medical and cell biologists working on cancer and, particularly, on HER2-overexpression cancers.

---

## [Referee Report · Reviewer #1 (Public review)]

Summary:

Soo-Yeon Hwang et al. synthesized and characterized a new set of Chalcone- and Pyrazoline-derived molecules targeting the interaction between ELF3, a transcription factor, and MED23, a coactivator for HER2 transcription. The authors employed biochemical analysis, cell-based assays, and an in vivo xenograft model to demonstrate that the lead compound, Compound 10, inhibits HER2 transcription and protein expression, subsequently inducing anticancer activity in gastric cancer models, particularly in trastuzumab-resistant cell lines. The obtained data is robust and supports the potential anticancer efficacy of Compound 10 for HER2+ gastric cancer.

Strengths:

The current manuscript proposes an alternative strategy for targeting HER2-overexpressing cancers by reducing HER2 transcription levels. The study presents compelling evidence that the lead compound, Compound 10, disrupts the binding of ELF3 to MED23, thereby inhibiting HER2 transcription. Notably, cell-based assays and xenograft models demonstrated the compound's significant antitumor activity in gastric cancer models.

---

## [Referee Report · Reviewer #2 (Public review)]

Summary:

The findings highlight the importance of targeting the ELF3-MED23 protein-protein interaction (PPI) as a potential therapeutic strategy for HER2-overexpressing cancers, notably gastric cancers, as an alternative to trastuzumab. The evidence, including the strong potency of compound 10 in inhibiting ELF3-MED23 PPI, its capacity to lower HER2 levels, induce apoptosis, and impede proliferation both in laboratory settings and animal models, indicates that compound 10 holds promise as a novel therapeutic option, even for cases resistant to trastuzumab treatment.

Strengths:

The experiments conducted are robust and diverse enough to address the hypothesis posed.

---

## [Referee Report · Reviewer #3 (Public review)]

Summary:

The authors synthesized a compound which can inhibit ELF3 and MED23 interaction which leads to inhibition of HER2 expression in gastric cancer.

Strengths:

Enough evidence shows the potency of compound 10 in inhibiting ELF3 and MED23 interaction.

---

## [Author Response]

The following is the authors’ response to the original reviews.

**Reveiwer#1 (Public Review):**
Weaknesses:While the novel compound showed a promising potency to the HER2-positive gastric cancer cells and xenograft model, it would be great to also to be evaluated with the HER2-positive breast cancer cell models. The author did not compare the current compounds with other therapeutic strategies targeting HER2 expression at the genetic level. It is unclear whether the EGFR inhibitors gefitinib and canertinib but not HER2-specific inhibitors (i.e. tucatinib) were used as a control in the manuscript.

We appreciate the reviewer’s insightful comments. Evaluating compound **10** on HER2-positive breast cancer cells is indeed crucial, especially given the established HER2-targeting therapies for breast cancer. In response to this concern, we conducted additional experiments to investigate the impact of compound 10 on HER2-positive breast cancer cell lines AU565 and BT474, specifically assessing its HER2 downregulating activity (Author response image 1).

**Author response image 1. sa4fig1:** HER2 downregulatory effect of compound 10 in HER2-positive breast cancer cell lines, AU565 and BT474.

The selection of gefitinib (an EGFR tyrosine kinase inhibitor) and canertinib (a pan-HER inhibitor) as positive controls in our manuscript is based on their demonstrated ability to inhibit the protein-protein interaction (PPI) between ELF3 and MED23, as previously reported J Adv Res. 47, (2023) 173-87. 10.1016/j.jare.2022.08.003; Cancer letters. 325, (2012) 72-9. 10.1016/j.canlet.2012.06.004. In referenced studies, SEAP reporter gene assay was utilized to screen compounds for their capacity to disrupt the ELF3-MED23 PPI. This assay involves GAL4-ELF3 binding to a GAL4 binding site in the SEAP reporter gene, followed by interaction with MED23, leading to RNA polymerase II recruitment and SEAP expression in cells J Am Chem Soc. 2004, 126(49), 15940. doi: 10.1021/ja0445140. Canertinib exhibited stronger inhibitory activity against ELF3-MED23 PPI compared to gefitinib, but also showed non-specific cytotoxicity. YK1 was subsequently developed based on structural analysis of the interfaces between gefitinib and MED23, and between ELF3 and MED23. Considering the previously validated inhibitory activities of gefitinib and canertinib, these drugs were selected as positive controls in the current study to compare the ELF3-MED23 inhibitory efficacy of novel compounds.

**Reveiwer#1 (Recommendations For the Authors):**
(1) It is unclear how compound 5 did not inhibit HER2 overexpression at mRNA but at protein levels as compounds 3 and 10. Could the author further explain the potential mechanism for compound 5?

While the exact mechanism remains unclear, the results indicated that compound 5 likely affects the protein level of HER2 through somewhat non-specific mechanisms rather than by inhibiting the ELF3-MED23 PPI. Based on this assessment, compound 5 was excluded from further investigation.

(2) The HER2 expression and its downstream signaling pathway assay are unclear about the approach. It needs to be included in the methods or supplementary.

We investigated the ELF3-MED23 PPI inhibitory activity and its subsequent effect on HER2 downregulation using a comprehensive approach involving multiple techniques to ensure precise and unbiased experimental results.

To assess PPI inhibition, we employed the following assays:

· SEAP reporter gene assay

· Fluorescence polarization (FP)

· Split-luciferase complementation assay

· GST-pulldown

· Immunoprecipiation (IP)

HER2 expression levels were evaluated through:

· SEAP reporter gene assay

· Luciferase promoter assay

· Quantification of HER2 mRNA using qPCR

· Measurement of HER2 protein levels via western blot analysis

To evaluate downstream signaling of HER2, we analyzed:

· Phosphorylation levels of MAPK (pMAPK) and AKT (pAKT)

These methods were systematically applied to elucidate the mechanism of action of compound **10** in inhibiting ELF3-MED23 interaction and subsequently downregulating HER2.

For clarity, we have revised the manuscript to provide a detailed description of the experimental methods to assess PPI, as described below.

“SEAP assay was performed as previously described to measure ELF3-MED23 PPI-dependent HER2 transcription [29]. In this assay, the GAL4-ELF3 fusion protein binds to one of the five GAL4 binding sites on the reporter gene (pG4IL2SX). The interaction between the GAL4-ELF3 fusion protein and endogenous MED23 induces the expression of the SEAP. Once expressed, SEAP acts as a phosphatase on the substrate 4-MUP (4-methyl umbelliferyl phosphate), resulting in increased fluorescence. The mammalian expression vector, …”

“FP assay was conducted following a previously described method to evaluate the molecular interaction between ELF3 and MED23 [29]. The FP assay operates on the principle of the molecular rotation dynamics. When a fluorescently labeled small molecule is excited by polarized light, the emitted fluorescence can be polarized or depolarized depending on the molecular status. Free small molecules rotate rapidly, altering the orientation of their fluorescence dipole and emitting depolarized light. However, when these small molecules bind to large molecules, such as proteins, the resulting complex rotates more slowly, and the emitted light retains much of its original polarization. In this study, different concentrations of (His)6-MED23391–582, as the large molecule, and 10 nM of FITC-labeled ELF3129–145 peptide, as the fluorescence-labeled small molecule, were combined in …”

(3) It is confusing to me about the order of the experiments, in which the SAR work came after the synthesis and a series of biochemical studies for the characterization of the synthetic compounds. What is the specific reason for this order?

We concluded that the current approach is appropriate because the analysis was not intended for structural modification and optimization through SAR (Structure-Activity Relationship) analysis. Instead, the primary objective was to elucidate the structural basis underlying the efficacy of PPI inhibition among compounds sharing the same scaffold. We believe this will provide valuable insights for future design and synthesis of new compounds.

(4) The yield for each step of the general synthesis needs to be included in the scheme 1.

Scheme 1 has been updated to include the yield of each step of the synthesis process.

(5) In line 532, the authors stated 28 compounds, should it be 26?

‘Twenty-eight compounds’ includes 26 newly synthesized compounds and 2 positive controls, gefitinib and canertinib.

(6) Introduction part, lines 74 to 75, "While HER2 gene amplification is the primary mechanism responsible for HER2 overexpression" may not be confirmed in lung cancers.

HER2 overexpression is usually a direct consequence of gene amplification, although overexpression can occur by other mechanisms [Nat Rev Cancer. 2009;9:463–475. doi: 10.1038/nrc2656.; Cell. 2007;129:1275–1286. doi: 10.1016/j.cell.2007.04.034.]. The levels of HER2 protein expression and gene amplification are linearly associated and highly concordant in breast cancer, colorectal cancer, ovarian cancer, and esophageal adenocarcinoma [World J Gastrointest Oncol. 2019, 11(4): 335–347. doi: 10.4251/wjgo.v11.i4.335; J Clin Oncol. 2002;20:719–26. doi.org/10.1200/JCO.2002.20.3.71; Oncology. 2001;61(Suppl 2):14–21. doi.org/10.1159/000055397; Science. 1989, 244(4905):707-12. doi: 10.1126/science.2470152; Cancer. 2014 Feb 1; 120(3): 415–424. doi: 10.1002/cncr.28435]. As reviewer mentioned, the linear association between of HER2 protein expression and gene amplification has not been fully established for NSCLC [ESMO Open. 2022, 100395. doi: 10.1016/j.esmoop.2022.100395].

Therefore, we change the sentence as describe below.

“While HER2 gene amplification is the primary mechanism responsible for HER2 overexpression in most HER2-positive cancers, except in lung cancer [16], high transcription rates of HER2 per gene copy have also been observed to contribute.”

(7) The abstract part, lines 31 and 32, the detailed experimental data for SEAP needs to be expressed in another way.

SEAP is a type of reporter gene assay. We revised the manuscript as follows and we additionally described it method part.

“Upon systematic analysis, candidate compound **10** was selected due to its potency in downregulating reporter gene activity of HER2 promoter confirmed by SEAP activity and its effect on HER2 protein and mRNA levels.”

(8) The author should combine the box for Chalcone, pyrazoline, Licochalcone E, and YK-1, Figures 1 and 2 into a new single Figure.

We revised the manuscript following the reviewer's comments.

(9) Provide the list of antibodies and sources for the cell-based and western blot assays.

Table S1 presents detailed information about the antibodies and dilution ratios used in the cell-based and western blot assays.

**Reveiwer#2 (Public Reviews):**
Weaknesses:The rationale behind the proposed structural modifications for the three groups of compounds is not clear.
**Reveiwer#2 (Recommendations For the Authors):**
(1) Based on previous work experience, it would be interesting to evaluate the in silico mode of interaction of compound **10**.

As suggested by the reviewers, we additionally performed *in silico* docking study to identify the mode of interaction of compound 10 (Author response image 2). As shown below, the results indicate that compound 10 shares a similar binding orientation with YK1, forming an H-bond with the H449 residue. Although it does not interact with the D400 residue, it was predicted to create an additional H-bond with S450, which is right next to H449, thereby reinforcing the overall binding of compound 10 to MED23. Moreover, compound 10 was additionally predicted to form a pi-pi interaction with F399, which has been previously identified as an important interaction for compounds to demonstrate outstanding PPI inhibitory effect against ELF3 and MED23.

**Author response image 2. sa4fig2:** Docking analysis of compound 10.

(2) The chalcones presented in this study are structurally similar to those previously presented by the group (ref 29). In said work, most of the compounds exhibited activities with IC50 values between 1.3 and 3 μM, with inhibition values at 10 μM ranging between 80 and 90% in the SEAP assay. These results are similar to those observed in this paper for the same assay. Can an explanation be found?

Chalcones are inherently flexible molecules, giving them a high chance of occupying critical hotspot residues within the binding interface of ELF3-MED23, irrespective of the side chains introduced to this moiety. However, depending on the type of side chains introduced, the overall drug-like properties of compounds can be significantly altered, while still maintaining their PPI inhibitory effect. The significance of this study lies in our effort to enhance metabolic stability through extensive introduction of methoxy groups and other hydrophobic side chains to the chalcone skeleton, while preserving high PPI inhibitory activity.

(3) Is the replacement of H and OH by OMe necessary? Does it improve any property (activity, selectivity, bioavailability, solubility, etc.)? Regarding the derivatives of group 2, why did they decide to replace the O-H, which in silico demonstrated favorable hydrogen bond interactions with Asp400? How do these molecules look in the binding site? Perhaps this is a point to discuss since the substitution of OH led to the obtaining of inactive molecules, or is the effect due to substitution with the terminal aromatic ring with 3 OMe?

We modified the hydroxyl group moiety of YK-1 into a methoxy group to reduce the polarity of the compound, thereby enhancing its cell membrane permeability (Author response image 3) and reducing the likelihood of rapid elimination through phase II metabolic pathways in vivo. Additionally, we considered the potential conversion of the methoxy group back to a hydroxyl group via phase I metabolism in vivo.

**Author response image 3. sa4fig3:** Impact of methoxy group introduction on TPSA (total polar surface area) of each molecule. TPSA of each molecule containing chalcone structure were calculated using the Molinspiration webserver.

(4) Lines 134 and 134: "Only compounds are in red."

We revised the manuscript following the reviewer's comments.

(5) Line 171: "Chalcone skeleton, shown in red."

We revised the manuscript following the reviewer's comments.

(6) Line 350: "N-1-acetyl-4,5-dihydropyrazoline."

We revised the manuscript following the reviewer's comments.

(7) Scheme 1. Replace "h" with "hr".

We revised the manuscript following the reviewer's comments. Scheme 1 has been replaced by a new version.

(8) Where is "Table S1" in SI?

Tables S1 and S2 are supposed to be included in SI. We will ensure that Tables S1 and S2 are properly uploaded to the SI section.

(9) In Figure 6, Graph D, to enhance comprehension, please incorporate red arrows indicating drug administration.

We revised Figure 6 (D) following the reviewer's comments. Red arrows indicating drug administration have been incorporated, along with a descriptive comment "Drug administration" next to each arrow. Additionally, the figure legend now includes a clear description of these additions.

**Reveiwer#3 (Public review):**
Weaknesses:Compound 10 potency as PPI inhibitor has been shown in only one cell line NCI-N87.
**Reveiwer#3 (Recommendations For the Authors):**
(1) The authors should show this compound 10 is effective in other gastric cancer cells like KATOIII, SNU1.

We evaluated the HER2 downregulating activity of compound 10 in the gastric cancer cell line, SNU216, which is confirmed to express high level of HER2 protein (Author response image 4).

**Author response image 4. sa4fig4:** HER2 downregulatory effect of compound 10 in HER2-positive gastric cancer cell line, SNU216. (A) Expression levels of HER2 and ELF3 in various gastric cancer cell lines. (B) HER2 downregulation in the SNU216 cell line following treatment with compound 10.